# *Campylobacter* vaccination reduces diarrheal disease and infant growth stunting among rhesus macaques

Sara M. Hendrickson [1], Archana Thomas[1], Hans-Peter Raué[1], Kamm Prongay[2], Andrew J. Haertel [2], Nicholas S. Rhoades[3], Jacob F. Slifka[1], Lina Gao [4], Benjamin K. Quintel[5], Ian J. Amanna [5], Ilhem Messaoudi [3] & Mark K. Slifka [1]✉

*Campylobacter*-associated enteric disease is estimated to be responsible for more than 160 million cases of gastroenteritis each year and is linked to growth stunting of infants living under conditions of poor sanitation and hygiene. Here, we examine naturally occurring *Campylobacter*-associated diarrhea among rhesus macaques as a model to determine if vaccination could reduce severe diarrheal disease and infant growth stunting. Compared to unvaccinated controls, there are no *Campylobacter* diarrhea-associated deaths observed among vaccinated infant macaques and all-cause diarrhea-associated infant mortality is decreased by 76% ($P = 0.03$). By 9 months of age, there is a 1.3 cm increase in dorsal length that equaled a significant 1.28 LAZ (Length-for-Age Z score) improvement in linear growth among vaccinated infants compared to their unvaccinated counterparts ($P = 0.001$). In this work, we show that *Campylobacter* vaccination not only reduces diarrheal disease but also potentially serves as an effective intervention that improves infant growth trajectories.

The 2019 Global Burden of Disease (GBD) study estimated that over 500,000 children under 5 years of age died of diarrheal disease in that year alone[1]. In addition to diarrhea-associated mortality, acute episodes of diarrhea as well as chronic or persistent diarrhea are known to increase the risk of infant growth stunting[2–5]. Growth stunting may also occur in the absence of overt diarrhea and in these cases, the underlying cause is believed to be due to a poorly understood condition described as Environmental Enteric Dysfunction (EED)[6]. EED is characterized by subclinical enteric pathogen infection with concomitantly increased gut inflammation, reduced nutritional uptake, and poor infant growth rates[4,6–11]. In 2010, it was estimated that *Campylobacter* spp. were associated with over 160 million cases

of foodborne illness and greater than 37,000 deaths[12]. *Campylobacter* spp. have been identified globally as a leading cause of childhood diarrhea with nearly 48 million cases per year among children less than 5 years of age[12]. Both symptomatic as well as asymptomatic *Campylobacter* infection appear to be important contributors to infant growth stunting[13–15]. In one study, children at 24 months of age with *Campylobacter* identified at every tested time point were 0.83 cm shorter ($P = 0.01$) than children with no *Campylobacter* and this was equivalent to a mean difference of 0.27 LAZ (length-for-age Z score)[16]. This suggests that interventions that reduce *Campylobacter*-associated disease could have a meaningful impact on infant growth velocity.

[1]Division of Neuroscience, Oregon National Primate Research Center, Oregon Health & Science University, Beaverton, OR 97006, USA. [2]Division of Animal Resources and Research Support, Oregon National Primate Research Center, Oregon Health & Science University, Beaverton, OR 97006, USA. [3]Department of Microbiology, Immunology and Molecular Genetics, University of Kentucky, College of Medicine, Lexington, KY 40506, USA. [4]Biostatistics and Bioinformatics Core, Oregon National Primate Research Center, Biostatistics Shared Resource, Knight Cancer Institute, Portland, OR 97239, USA. [5]Najít Technologies, Inc., Beaverton, OR 97006, USA. ✉e-mail: slifkam@ohsu.edu

To study diarrheal disease under conditions that mimic key aspects of growth stunted children living under conditions of poor sanitation and hygiene, we utilized an outdoor-housed rhesus macaque model of natural enteric infection. Despite access to clean water and a well-balanced nutrient-rich diet, rhesus macaques (RM) have dysbiotic gut microbiomes that resemble the those of people living in resource-poor settings such as Amerindians from Venezuela, as well as those living in Malawi, Burkina Faso, or Bangladeshi slums[17–19]. At the Oregon National Primate Research Center (ONPRC), infant macaques are exposed to a number of enteric pathogens with substantial infection rates for *Campylobacter* (78%), *Shigella* (26%), *Cryptosporidium* (7%) and/or *Giardia* (33%) observed by 1 month of age[20]. In the absence of clinically overt diarrhea, RM infants develop histological abnormalities of the small intestine by 8–11 months of age that are consistent with the histological findings of infants/small children with EED[21] including villous blunting, decreased crypt-to-villus ratio, and local inflammation[20]. Interestingly, poor infant growth trajectories and low serum tryptophan levels (a measure of malabsorption, altered metabolism, or possibly colonic leakage) correlated with histopathology of the large intestine instead of the small intestine. These results led to the hypothesis that a mechanistic link between poor gut health and reduced infant growth trajectories may be due, at least in part, to the colon/large intestine functioning as an energy salvage organ[22] when the small intestine is compromised[20]. Diarrheal disease is common among outdoor-housed non-human primates (NHP)[23–27] and similar to humans, the diarrhea-associated mortality rate of infants is significantly higher than that observed among adults[23]. We[27] and others[28–32] have developed a number of experimental vaccines against *Campylobacter*. However, the mammalian studies have been performed mainly with adult animals or adult human subjects and there is currently little to no information regarding how *Campylobacter* vaccination might specifically impact the health and growth kinetics of infants raised in a high-exposure, *Campylobacter*-endemic environment.

In this work, we compared three different *Campylobacter* vaccination schedules including a group of infant macaques in which the dams were vaccinated during pregnancy followed by vaccination of their offspring. In each case, *Campylobacter* vaccination of infant macaques elicited antibacterial serum antibody responses that were higher than that achieved after natural *Campylobacter* infection alone. We found that the rates of *Campylobacter*-associated diarrhea were reduced among vaccinated infants, with no *Campylobacter*-associated deaths recorded (0/90 vs. 7/248 *C. coli* deaths among unvaccinated infants). All-cause diarrhea-associated mortality was also reduced by 76% (2/90 all-cause deaths among vaccinated infants vs. 23/248 all-cause diarrhea deaths among unvaccinated infants, $P = 0.03$). In addition to lower diarrhea-associated morbidity and mortality, *Campylobacter* vaccination of both dams and infants also resulted in significantly improved infant growth kinetics. Together, these results indicate that in an environment with high and continuous enteropathogen exposure, *Campylobacter* vaccination may be a viable new approach for decreasing the burden of diarrheal disease while significantly improving infant growth trajectories.

## Results

### *Campylobacter* vaccination does not impact microbiome maturation during infancy

We performed *Campylobacter* vaccination studies among outdoor-housed RM living in small shelter groups that are prone to high rates of enteric disease[20,23,27]. The study involved infant macaques born in 2017 and 2018 and both LOS (lipopolysaccharide) and CPS (capsular polysaccharide) loci of the circulating strains of *C. coli* sequenced in this time frame (2015–2018) did not match the *C. coli* vaccine strain (isolated in 2013) and therefore these experiments represent a robust heterologous challenge model for measuring vaccine-mediated

protection against *Campylobacter*-associated enteric disease[27]. In terms of study design, we examined 3 groups of vaccinated animals plus unvaccinated control animals (i.e., Controls) for a total of 4 groups of infant macaques. One vaccine group was comprised of vaccinated dams and their infants (VDI: Vaccinated Dams and Infants) in which the dams were vaccinated twice with a 40 µg dose of $H_2O_2$-Campy$_c$ during pregnancy and their infants were vaccinated with the same dose at 1, 3, and 12 months of age (VDI-1/3/12). The other groups of vaccinated infants born to unvaccinated dams (VI: Vaccinated Infants) were immunized on a 1, 3, 12-month schedule or on a 1, 3, 5-month vaccination schedule (VI-1/3/12 and VI-1/3/5, respectively). Serum antibody responses were monitored for up to 12 months of age, and the incidence of diarrhea was passively monitored for up to 18 months using an electronic health database that tracks diarrheal disease among the NHP at the ONPRC. Across all RM age groups, *Campylobacter coli* (*C. coli*) accounts for 59% of hospitalized cases of diarrhea, followed by *Shigella* (12%) and *C. jejuni* (5.9%)[27]. Prior studies indicated that ~80% of infants are colonized with *Campylobacter* spp. by one month of age with between 69% to 97% of juveniles and adults in the same breeding groups also showing asymptomatic carriage at any given point in time[27]. Similarly, we found that 21/25 (84%) of unvaccinated infant Controls, 19/25 (76%) of VDI-1/3/12 infants, 22/28 (79%) of VI-1/3/12 infants, and 43/50 (86%) of VI-1/3/5 infants were *C. coli* culture-positive by 1 month of age (Table 1). The colonization status of the vaccinated dams did not influence the rates of *C. coli* colonization of their infants since 76% of the infants born to vaccinated dams in the VDI-1/3/12 group were colonized with *C. coli* by 1 month of age and this was not significantly different than the frequency of *C. coli*-positive infants (79%) in the VI-1/3/12 group that were born to unvaccinated dams in the same birth year ($P = 0.99$, Fisher's Exact Test). Moreover, of the 6 infants in the VDI-1/3/12 group that were *C. coli*-negative at 1 month of age, 2 infants were born to *C. coli*-negative dams and 4 were born to *C. coli*-positive dams, indicating that there was no direct relationship between the colonization status of the vaccinated dams and their infants at 1 month post-delivery. In these studies, being *C. coli*-negative at 1 month of age did not appear to predict future risk of *C. coli* diarrhea since 4/21 (19%) of the Control infants that were *C. coli*-positive at 1 month of age were subsequently hospitalized with *C. coli* diarrhea whereas 2/4 (50%) of Control infants that were *C. coli*-negative at 1 month also were eventually hospitalized with *C. coli* diarrhea. At 1 month of age, approximately 4–8% of the infants were infected with *C. jejuni* and 0–6% were infected with *Shigella*, with cumulative prevalence of these enteric pathogens increasing over the course of the study (Table 1). The 1 month colonization data indicates that these infant groups were similar in terms of their initial pre-existing *C. coli* carriage rates at the time of first vaccination and illustrates one of the challenges of this enteric disease model since the gut microbiome of the majority of infant macaques are colonized with *Campylobacter* prior to the first opportunity to directly administer a preventative vaccine.

The impact of vaccination against *C. coli* on the development of the gut microbiome among infant macaques was determined using 16 S rRNA gene amplicon sequencing (Fig. 1). Since diarrheal disease has been shown to significantly impact the development of the infant gut microbiome[19], this analysis was conducted on samples from animals that remained asymptomatic during the study period. Longitudinal development of microbial diversity differed between the unvaccinated Controls examined in 2017 vs. the unvaccinated Controls in 2018 with samples collected at the 6- and 9-month timepoints being significantly more diverse in 2017 than 2018 (Mixed effect model on cohort, $P = 0.0000000016$, Šídák's post-hoc $P = 0.008$ at 6 months and Šídák's post-hoc $P = 0.002$ at 9 months, Fig. 1a). To ensure that this yearly variability did not affect the ability to determine the impact of vaccination on the microbiome, all comparisons between vaccinated and unvaccinated infants were made using their contemporaneous

**Table 1 | Burden of C. coli, C. jejuni, and Shigella spp. colonization of rhesus macaques**

| Microbe | Control Infants, % (n) | | VDI-1/3/12 Infants, % (n) | | VI-1/3/12 Infants, % (n) | | VI-1/3/5 Infants, % (n) | | Unvaccinated Dams, % (n) | | Vaccinated Dams, % (n) | |
|---|---|---|---|---|---|---|---|---|---|---|---|---|
| | 1 Month | Cumulative | 1 Month | Cumulative | 1 Month | Cumulative | 1 Month | Cumulative | 1 Month | Cumulative | 1 Month | Cumulative |
| C. coli | 84% (21) | 100% (25) | 76% (19) | 100% (24)* | 79% (22) | 100% (27)* | 86% (43) | 100% (50) | 73% (57) | 79% (62) | 56% (14) | 100% (25) |
| C. jejuni | 4% (1) | 50% (15) | 8% (2) | 17% (4)* | 4% (1) | 26% (7)* | 8% (4) | 58% (29) | 4% (3) | 4% (3) | 8% (2) | 20% (5) |
| Shigella | 0% (0) | 20% (11) | 0% (0) | 33% (8)* | 0% (0) | 30% (8)* | 6% (3) | 22% (11) | 6% (5) | 12% (9) | 16% (4) | 56% (14) |

Prevalence of C. coli, C. jejuni and Shigella spp. colonization was determined at 1 month of age and also cumulatively (1–12 months of age) using microbial culture of rectal swabs. C. coli colonization occurs at an early age with 96% (24/25), 96% (23/24), 96% (26/27), and 96% (48/50) of the infant macaques in the Control, VDI-1/3/12, VI-1/3/12, or VI-1/3/5 cohorts colonized with C. coli by 3 months of age. Infant Controls include all enrolled control animals from 2017 to 2018 study cohorts with available longitudinal culture data and animals were sampled up to 6 times, at 1, 3, 5, 6, 9, and 12-month time points. The animals in the 2017 infant groups, VDI-1/3/12 and VI-1/3/12, were sampled up to 5 times, at 1, 3, 6, 9, and 12-month time points. Animals in the 2018 infant group, VI-1/3/5, were sampled up to 6 times, at 1, 3, 5, 6, 9, and 12-month time points. Vaccinated dams (n = 78) were sampled up to two times, at 1 and 3 months after giving birth. *For the VDI-1/3/12 group, there were 25 infants examined at 1 month of age but only 24 infants were tested for cumulative incidence of enteric pathogens due to humane euthanasia of 1 infant for non-diarrhea associated causes before 3 months of age. Likewise, for the VI-1/3/12 group, there were 28 infants examined at 1 month of age but only 27 infants were tested for cumulative incidence of enteric pathogens due to humane euthanasia of 1 infant for non-diarrhea associated causes before 3 months of age.

birth cohort controls. No difference was observed in terms of microbial alpha diversity, as measured by observed amplicon sequence variant (ASVs), at any timepoint after vaccination between the 2017 vaccinated groups (VDI-1/3/12 and VI-1/3/12) and their corresponding unvaccinated 2017 Controls or the 2018 vaccinated group (VI-1/3/5) and their contemporaneous unvaccinated 2018 Controls (Mixed effect model on 2017 cohort, unadjusted $P = 0.26$ and on 2018 cohort, unadjusted $P = 0.57$, Fig. 1a). Additionally, the overall community composition (i.e., beta diversity), calculated as weighted UniFrac distance, did not differ between vaccinated groups and their contemporaneous Controls at any timepoint (Fig. 1b, c). In agreement with these results, vaccination did not lead to any differentially abundant taxa in the infant gut microbiome at any time point, and there were no differences in richness or composition of the dam gut microbiome associated with vaccination (One-way ANOVA, $P = 0.48$). Overall, these data indicate that while microbial diversity can differ from one year to the next, vaccination against C. coli does not disrupt the development of the infant macaque gut microbiome.

### Longitudinal analysis of Campylobacter-specific antibody responses

To determine the immunogenicity elicited by each of the different Campylobacter immunization schedules, total anti-Campylobacter serum IgG responses were measured longitudinally among vaccinated infants with no history of C. coli diarrhea in comparison with unvaccinated Controls (Fig. 2a). The antibody titers shown at month = 0 (marked M: maternal) represent the corresponding maternal anti-C. coli antibody titers of the dams from serum samples obtained when their infants had samples drawn at 1 month of age. All of the unvaccinated dams of Control infants were seropositive for C. coli (geometric mean titer: 3990 ELISA units; EU), as might be expected following years of endemic exposure to Campylobacter spp. Likewise, nearly all Control infants had detectable anti-Campylobacter antibodies at 1 month of age. This may be the result of transferred maternal antibodies, endogenous production of antibacterial antibodies, or a combination of these two mechanisms. Interestingly, anti-Campylobacter antibodies increased 10-fold from 1 month to 12 months of age (627 and 6730 EU, respectively) among unvaccinated Control infant macaques in the absence of clinically apparent C. coli diarrhea (Fig. 2b). These antibody titers were similar to those observed among the matched adult RM (3990 EU) but were not significantly different from unvaccinated Control infants that were hospitalized with C. coli-confirmed diarrhea (12 months-Dx; 5,590 EU, $P = 0.56$, Fig. 2b). Vaccination of dams during pregnancy in the VDI-1/3/12 group resulted in anti-Campylobacter titers of 11,100 EU (Fig. 2b). The antibacterial antibody levels of their offspring at 1 month of age (4560 EU) were at levels similar to the titers normally observed among naturally infected adult RM and were significantly higher than that observed among unvaccinated Control infants at 1 month of age (627 EU, $P = 0.002$, Fig. 2b). These high levels of pre-existing maternal antibodies did not inhibit infant vaccination since immunization at 1 and 3 months of age resulted in sustained anti-Campylobacter antibody titers of 34,300 EU at the 12-month time point. Vaccination of infants born to unvaccinated dams (VI-1/3/12) also resulted in high and sustained antibody titers at the 12-month time point (28,900 EU). Vaccinating infants three times before the first year of age (i.e., 1, 3, and 5 months of age; VI-1/3/5) provided a transient increase in antibody titers at 6 months (i.e., 1 month after the third dose; Fig. 2a) but by 12 months of age, the antibacterial antibody titers (34,500 EU) were not significantly different from those observed among infants that received only two doses of vaccine given at 1 and 3 months of age (34,300 EU, $P = 0.46$ and 28,900 EU, $P = 0.31$, respectively). Within each vaccine or control group, the Campylobacter-specific antibody titers among animals diagnosed with C. coli diarrhea (Dx) were not significantly higher compared to their non-diarrhea counterparts at 12 months of age,

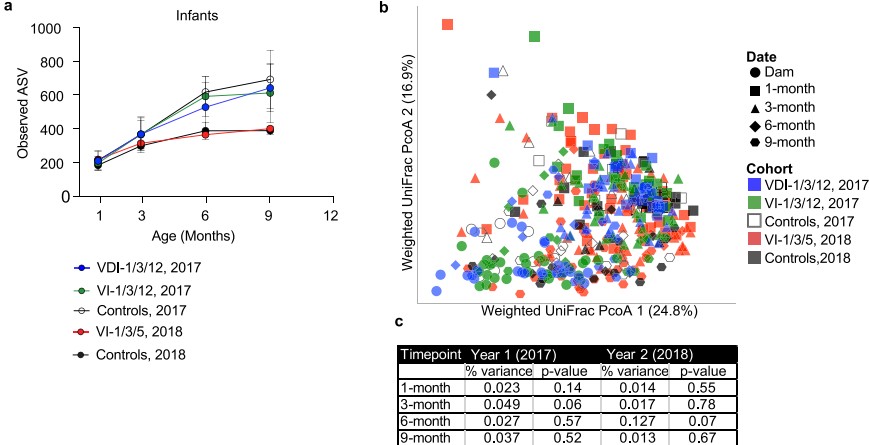

| Timepoint | Year 1 (2017) | | Year 2 (2018) | |
|---|---|---|---|---|
| | % variance | p-value | % variance | p-value |
| 1-month | 0.023 | 0.14 | 0.014 | 0.55 |
| 3-month | 0.049 | 0.06 | 0.017 | 0.78 |
| 6-month | 0.027 | 0.57 | 0.127 | 0.07 |
| 9-month | 0.037 | 0.52 | 0.013 | 0.67 |

**Fig. 1 | Vaccination against *C. coli* does not impact the maturation of the infant macaque gut microbiome.** The fecal microbiome was profiled using 16 S rRNA gene amplicon sequencing of samples collected longitudinally for all infants from two birth cohorts (2017 and 2018) to assess the development and maturation of their microbiome. **a** Longitudinal measurements of Observed Amplicon Sequencing Variants (ASV) were used to assess microbial richness and the lines are colored by vaccination cohort and year with each line indicating the average development for that group with symbols and error bars indicating the average ± SEM. Significance was assessed by a mixed effect model comparing the vaccine arms against controls across time within each cohort. In each case, there was no significant difference between these groups in both birth cohorts (*P* > 0.05). **b** Principal coordinate analysis (PcoA) Weighted unique fraction metric (UniFrac) distance, a dimensional reduction technique in which beta-diversity can be visualized in 2-dimensions with distance between points indicating dissimilarity of overall microbiome composition, are indicated by symbol (i.e., by time point) and colors indicate birth year and vaccination cohort. **c** A PERMANOVA result table was prepared to assess the impact of vaccination on microbiome composition at each timepoint, with split birth cohorts to account for yearly variation, and all vaccine arms considered separately. *P* values shown are unadjusted one-sided *F* test *P* values from each PERMANOVA test. Time points with non-significant test results indicate that the overall community composition cannot distinguish between the indicated groups. The number of microbiome samples available for analysis varied by group and by time point as follows: VDI-1/3/12 (*n* = 24, 23, 9, 9); VI-1/3/12 (*n* = 28, 25, 9, 10), 2017 Controls (*n* = 10, 10, 9, 9); VI-1/3/5 (*n* = 46, 45, 10, 42) and 2018 Controls (*n* = 15, 9, 9, 13) were examined at the 1, 3, 6, and 9 month time points, respectively. A single microbiome sample was obtained from VDI-1/3/12 dams (*n* = 25), VI-1/3/12 dams (*n* = 28), and 2017 Control dams (*n* = 8) at 1 month post-delivery. Source data are provided as a Source Data file.

indicating that in this endemic animal model, clinical disease does not necessarily result in higher antibacterial immune responses (Controls; *P* = 0.60, VDI-1/3/12; *P* = 0.85, VI-1/3/12; *P* = 0.31, and VI-1/3/5; *P* = 0.89, for non-diarrheal vs. diarrheal animals in each group, respectively, two-sided student's *t* test with unequal variance, Fig. 2b). Similar results were observed when anti-*Campylobacter* flagellin antibodies were measured (Supplementary Fig. 1). In addition to measuring serum antibody titers, milk was collected from a subset of dams in each group when their infants were at 1 and 3 months of age and these samples were tested for *Campylobacter*-specific antibodies (Supplementary Fig. 2). *Campylobacter*-specific IgG was below the limits of detection in the milk whereas *Campylobacter*-specific IgA levels reached geometric mean titers of 195 EU, 102 EU, and 113 EU, at the 1 month time point for the VDI-1/3/12, VI-1/3/12, and Control groups, respectively before declining to nearly background levels of antibody equivalent to approximately 29 EU, 24 EU, and 15 EU at the 3 month time point for the VDI-1/3/12, VI-1/3/12, and Control groups, respectively. There was no significant difference in *Campylobacter*-specific IgA titers between the milk from vaccinated dams in the VDI-1/3/12 group and the other two groups of unvaccinated dams (VI-1/3/12 dams, *P* = 0.93 at 1 month and *P* = 0.69 at 3 months; or Control dams, *P* = 0.66 at 1 month and *P* = 0.64 at 3 months, respectively, unadjusted one-way ANOVA, Supplementary Fig. 2).

## *C. coli* diarrhea among vaccinated and unvaccinated infant macaques

Diarrheal disease incidence/hospitalization and mortality were monitored for up to 18 months of age and spanned across two birth cohorts born in 2017 and 2018 (Fig. 3). In 2017, a group of 139 unvaccinated Control animals were monitored for an average of 420 days, resulting in 58,326 exposure days in which there were 28 hospitalized cases of *C. coli* diarrhea and an incidence rate of 20%. Contemporaneous groups of vaccinated infants (VDI-1/3/12, *n* = 25 and VI-1/3/12, *n* = 28) were monitored in parallel for diarrheal disease. Infants in the VDI-1/3/12

group were monitored for an average of 402 days, resulting in 10,060 exposure days in which there were 2 cases of *C. coli* diarrhea and an incidence rate of 8%. Based on a time-to-event Cox proportional hazards model, this resulted in an estimated Vaccine Efficacy (VE) of 58% (*P* = 0.22) in the Intent-to-Treat cohort that included all animals with diarrhea cases that occurred at any age. A Per-Protocol subset was also examined and it was defined based on cases of *C. coli* diarrhea that occurred ≥28 days after booster vaccination at the 3-month time point. Based on this analysis, VE reached 79% (*P* = 0.09). Infants in the VI-1/3/12 group were monitored for an average of 359 days, totaling 10,045 exposure days in which there were 3 cases of *C. coli* diarrhea, resulting in an incidence rate of 11% and VE = 41% (*P* = 0.37) in the Intent-to-Treat cohort and VE = 41% (*P* = 0.37) in the Per-Protocol cohort. No cases of *C. coli* diarrhea were reported among either of these vaccinated infant cohorts during the combined 5242 exposure days that occurred after the third vaccination at 12 months of age (Fig. 3).

In 2018, a group of 127 unvaccinated Control animals were monitored for an average of 436 days, resulting in 55,426 exposure days in which 31 hospitalized cases of *C. coli* diarrhea were identified for a disease incidence of 24%. A contemporaneous group of vaccinated infants (VI-1/3/5, *n* = 50) were followed for an average of 449 days, resulting in 22,428 exposure days in which there were 8 cases of *C. coli* diarrhea, an incidence rate of 16%, and an estimated VE = 36% (*P* = 0.26) in the Intent-to-Treat cohort and VE = 53% (*P* = 0.08) in the Per-Protocol group. Since the *C. coli* diarrhea incidence rates among unvaccinated infants were similar across both the 2017 and 2018 birth cohorts (Supplementary Fig. 3), we combined the data from the vaccinated cohorts (all vaccinated infants, *n* = 103) and compared this to the combined data from the unvaccinated cohorts (all controls, *n* = 266). Overall, we found VE = 41% (*P* = 0.08) in the combined Intent-to-Treat vaccine group and this reached a statistically significant VE = 55% (*P* = 0.02) in the combined Per-Protocol infant vaccine group.

Prior work involving *Campylobacter* vaccination of mainly adult RM found VE = 83% (*P* = 0.048) against *C. coli* (i.e., the target of the

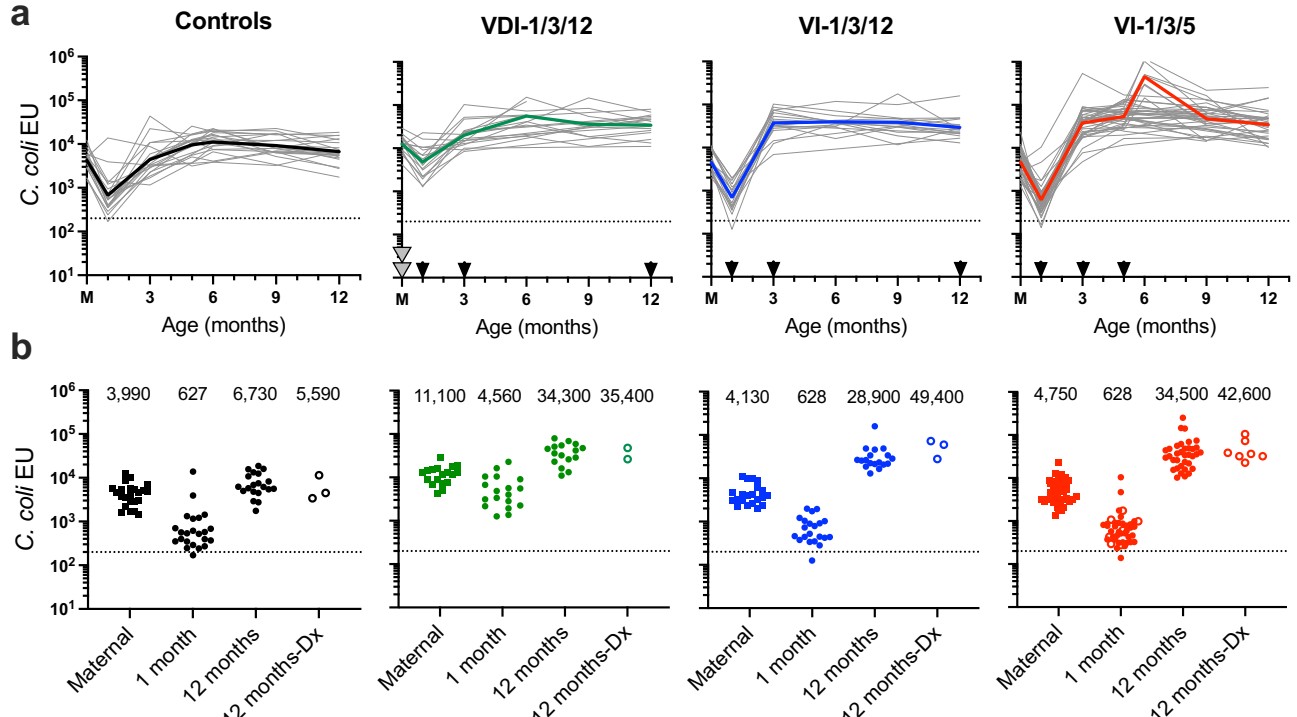

**Fig. 2 | Comparison of *Campylobacter*-specific antibodies elicited after natural infection or vaccination.** To measure antibacterial antibody responses against *C. coli*, serum samples were drawn at regular intervals from Control infants (1, 3, 6, 9, and 12 months), VDI-1/3/12 infants (1, 3, and 12 months), VI-1/3/12 infants (1, 3, and 12 months), and VI-1/3/5 infants (1, 3, 5, and 12 months, with a subset of 7 animals sampled at 6 months). **a** Among animals with no history of *C. coli* diarrhea, *C. coli* ELISA Units (EU) were measured longitudinally for each infant (thin lines) and the geometric mean was calculated at each time point (thick lines). The 0 month time point represents maternal (M) antibody titers from serum samples of each infant's dam that were drawn at the infant's 1 month time point. Control infants were not vaccinated whereas the other three groups of infants were vaccinated on a 1, 3, 12 month schedule or a 1, 3, 5 month schedule, as indicated by the black arrow heads on the *X*-axis. The gray arrow heads indicate that the dams in VDI-1/3/12 cohort were vaccinated twice during pregnancy. **b** Comparisons were made between maternal antibodies of the dams with their respective infants at the 1 month and 12 month time points among non-diarrheal animals within each group. In addition, a comparison was also made between 12 month time points from animals that were asymptomatically infected and/or vaccinated against *C. coli* (12 months) with infants from each group that were hospitalized for *C. coli* diarrhea (Dx) prior to the 12 month time point (12 months-Dx). No infants were diagnosed with diarrhea at 1 month of age but the open symbols at the 1 month time point represent animals that would eventually be diagnosed with *C. coli* diarrhea by the 12 month time point. The numbers above each group represent the geometric mean antibody titer. Source data are provided as a Source Data file.

vaccine) with a modest trend towards a lower frequency of all-cause diarrhea (VE = 69%, $P$ = 0.082)[27]. To determine if similar trends were observed following *Campylobacter* vaccination of infant macaques, we measured the incidence of all-cause diarrhea in the Per-Protocol population. Vaccine efficacy against all-cause diarrhea-associated hospitalization was 14% ($P$ = 0.71) for VDI-1/3/12, not estimable ($P$ = 0.61) for VI-1/3/12, and 54% ($P$ = 0.04) for VI-1/3/5 animals. After combining the data from all vaccinated infant macaque groups, VE = 27% ($P$ = 0.18). Of the 25 hospitalized cases of all-cause diarrhea among the vaccinated animals, 13 (52%) were *C. coli*-associated, followed by 2 cases of *C. jejuni* (8%), 1 case of *Shigella* (4%) and 9 cases (36%) of unknown/undetermined etiology. Among the unvaccinated control animals, there were 83 cases of all-cause diarrhea including 59 (71%) that were *C. coli*-associated, followed by 8 cases of *C. jejuni* (9.6%), 3 cases of *Shigella* (3.6%), 1 case of *C. jejuni* plus *Shigella* (1.2%), and 12 cases (14.5%) of unknown/undetermined etiology.

In addition to hospitalization of infant macaques due to severe diarrhea and dehydration, a proportion of animals may succumb to diarrheal disease or require humane euthanasia due to complications despite oral hydration, intravenous hydration, nutritional supplementation, antibiotic therapy, and expert supportive care provided by the attending veterinarians. To determine if *Campylobacter* vaccination might improve infant survival rates, we reviewed the medical records of vaccinated and unvaccinated infants through the first 18 months after birth (Fig. 4). In terms of *C. coli* diarrhea-associated

mortality, 7 lethal cases were identified among 248 unvaccinated Controls (2.8%) compared to 0 cases identified among 90 vaccinated infants, resulting in 100% VE ($P$ = 0.2, Fig. 4a). Although *C. coli*-specific deaths were relatively rare, when expanded to all-cause diarrhea-associated mortality, there were significant differences between vaccinated and unvaccinated animals. Among the 90 vaccinated infants, there were 2 diarrhea-associated deaths (2.2%) of either unknown etiology or in which only normal enteric bacteria were identified including 1 non-*C. coli* death in the VDI-1/3/12 group and 1 non-*C. coli* death in the VI-1/3/12 group. Among the 248 unvaccinated control infants, there were 23 diarrhea-associated deaths including 7 cases of *C. coli* (30%), 2 cases of *C. jejuni* (8.7%), 1 case of *Shigella* (4.3%), and 13 cases of unknown etiology (57%), resulting in an overall mortality rate of 9.3%. Comparison of these two groups indicated a significant impact of *Campylobacter* vaccination on infant mortality with an estimated VE = 76% ($P$ = 0.03, Fig. 4b). Together, this suggests that *Campylobacter* vaccination may not only protect against its respective vaccine target (*C. coli*), but may also improve gut health and provide resistance to potentially life-threatening diarrheal disease by other enteric pathogens that are not included in the vaccine formulation.

**Impact of *Campylobacter* vaccination on infant growth stunting**

Both symptomatic and asymptomatic *Campylobacter* infection are associated with human growth stunting[13] and infants with low/negative *Campylobacter* carriage have better growth trajectories compared to

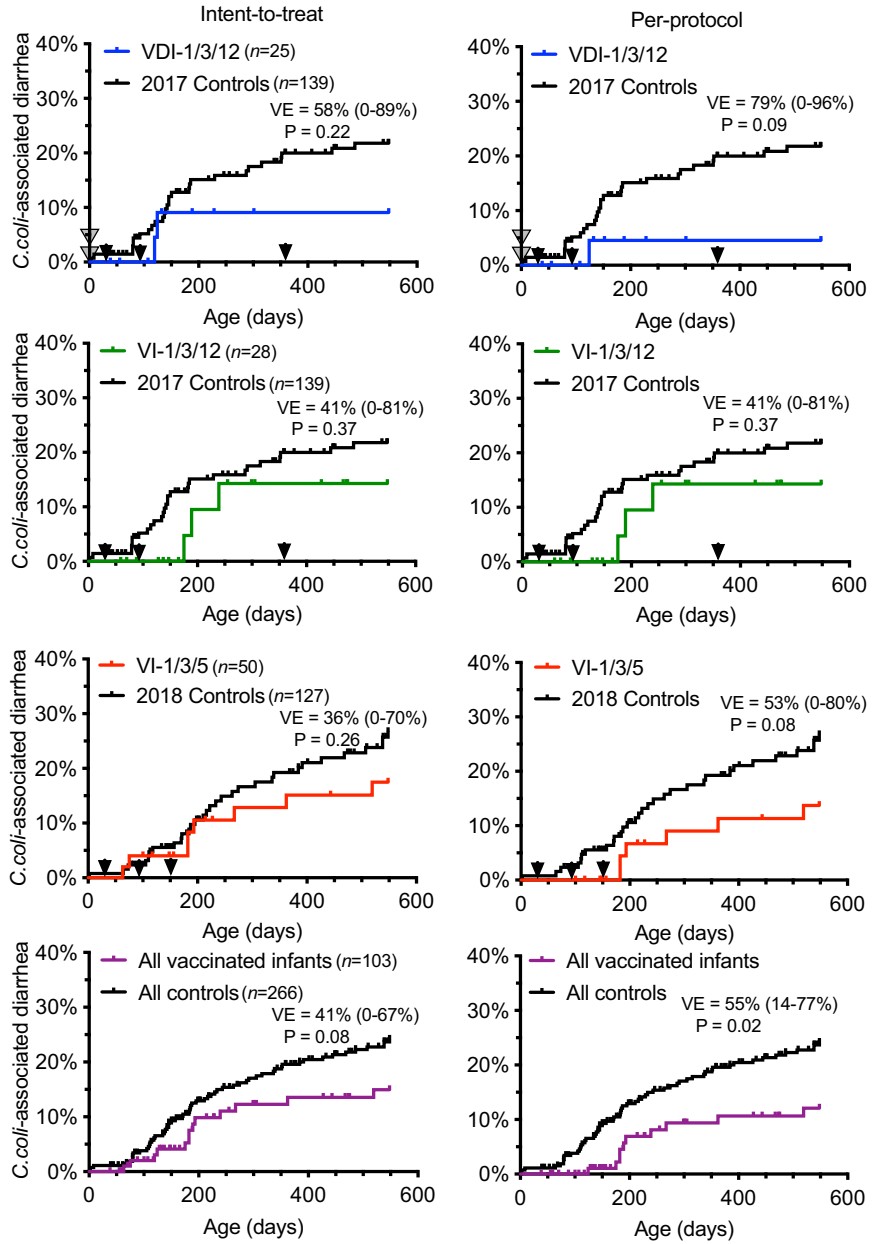

**Fig. 3 | Risk of *C. coli*-associated diarrhea among vaccinated infant macaques and naïve controls.** Vaccinated infant macaques were monitored for hospitalized cases of *C. coli* diarrhea in comparison to unvaccinated Controls for up to 18 months after birth. The study was performed across two birth cohorts with VDI-1/3/12 and VI-1/3/12 compared to unvaccinated Controls born in 2017 and the VI-1/3/5 cohort was compared to unvaccinated Controls born in 2018. Since the incidence of *C. coli* diarrhea was similar among unvaccinated animals across both years (Supplementary Fig. 2), a combined analysis of all animals from 2017 to 2018 was also performed. To measure vaccine efficacy (VE), the incidence of *C. coli* diarrhea was based on all hospitalized cases of diarrhea, regardless of time frame in relation to vaccination (Intent-to-treat) or *C. coli* diarrhea cases were counted beginning 28 days after booster vaccination administered at the 3 month time point (Per-protocol). *P* values were determined by Log-rank (Mantel–Cox Test) and a Cox PH hazard ratio was used for determining VE with no post-hoc adjustments. Source data are provided as a Source Data file.

infants with high *Campylobacter* carriage rates[16]. To determine if *Campylobacter* vaccination might improve infant growth trajectories, we examined dorsal length of a subset of 20 vaccinated infants (n = 10 VDI-1/3/12 and 10 VI-1/3/12 infants) vs. 10 unvaccinated Controls at 1, 3, 6, 9, and 12 months of age (Fig. 5). At 1 month of age, the size of the vaccinated and unvaccinated control groups was almost identical (34.2 cm vs. 34.1 cm, respectively, *P* = 0.85, student's *t* test) but during the following months the difference in linear growth became more apparent between the two groups. The overall improvement in length among the vaccinated animals represented approximately 0.5 cm, 0.8 cm, 1.3 cm, and 0.4 cm in additional dorsal length among the

vaccinated infants at 3, 6, 9, and 12 months of age, respectively. At 1 month of age, none of the animals had been hospitalized for diarrhea although one unvaccinated Control infant (marked with a red asterisk) would later be hospitalized with *C. coli* diarrhea at 79 days of age. At 3 months of age, none of the vaccinated infants had experienced clinical diarrhea but the one unvaccinated Control infant that was hospitalized with *C. coli*-associated diarrhea was found to be the second smallest animal in its group. These results appear to be similar to those observed among humans[2-5], wherein overt diarrhea may result in reduced infant growth rates. Indeed, by 6 months of age, this infant was the smallest animal among the entire cohort of both vaccinated

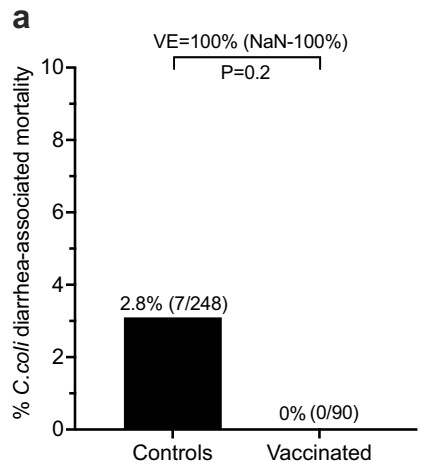
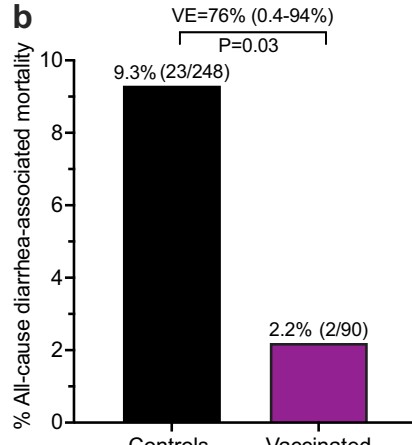

**Fig. 4 | Vaccine-mediated protection against *C. coli*-associated mortality and all-cause diarrhea-associated mortality.** Vaccinated infants (VDI-1/3/12, VI-1/3/12, and VI-1/3/5, *n* = 90) and unvaccinated Controls (*n* = 248) were monitored for *C. coli*-associated mortality (**a**) or all-cause diarrhea-associated mortality (**b**) for up to 18 months after birth. Vaccine efficacy (VE) was determined by the attack rate among unvaccinated infants minus the attack rate of the vaccinated infants, divided by the attack rate among the unvaccinated infants ×100 and are shown with 95% Taylor series confidence intervals. *P* values were calculated using two-sided Fisher's Exact Test. NaN not a number. Source data are provided as a Source Data file.

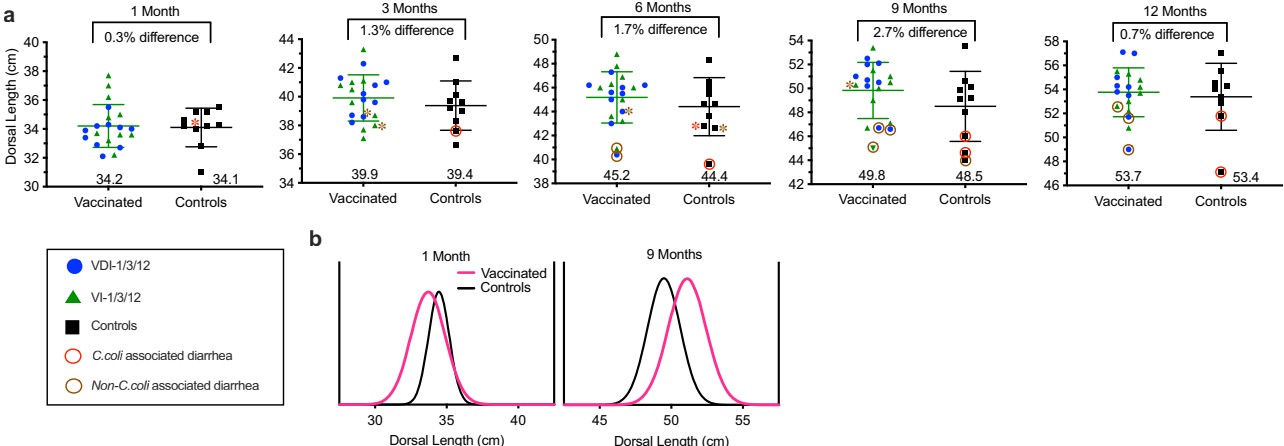

**Fig. 5 | Improved growth rate kinetics among infant macaques that received *Campylobacter* vaccination.** Infant dorsal length was measured longitudinally among a subset of Vaccinated infants and unvaccinated Controls. **a** At the indicated time points (1, 3, 6, 9, and 12 months) the mean dorsal length (±standard deviation) was measured in centimeters (cm). The percentage difference in linear growth between the vaccinated infants and unvaccinated Controls is provided at the top of each panel and the numbers at the bottom represent infant dorsal length in centimeters. Symbols with an asterisk (*) indicate animals that appeared healthy at the time of measurement but were hospitalized with diarrhea at the next following time point. Brown asterisks indicate future hospitalization with all-cause diarrhea and red asterisks indicate future hospitalization with *C. coli* diarrhea. The study began with 10 vaccinated infants/group (*n* = 10 VDI-1/3/12 infants with 9 remaining at both 9 and 12 months and *n* = 10 VI-1/3/12 infants with 9 remaining at 12 months) and 10 unvaccinated Controls (*n* = 10 Control infants with 9 remaining at 12 months). Brown circles indicate animals that had been hospitalized with all-cause diarrhea whereas red circles indicate animals that were hospitalized with *C. coli*-associated diarrhea. **b** Histogram analysis of dorsal length was performed at the 1 month (baseline) and 9 month time points for comparison with Controls. Black lines represent unvaccinated Controls and pink lines represent Vaccinated infants. Dorsal length histogram bin width was set at 0.5 cm for the 1 month dataset and 1.0 cm for the 9 month dataset. Source data are provided as a Source Data file.

and unvaccinated infants (Fig. 5a). Interestingly, several animals fell to the bottom of their peer groups for dorsal length (see asterisks) before they were eventually hospitalized for diarrhea, suggesting that these may be examples of EED that later progress to clinical diarrheal disease. At the 6-month time point, there were 2 cases of non-*C. coli* diarrhea identified among the Vaccinated cohort and, similar to the infant with *C. coli* diarrhea in the Control group, the growth trajectories of these diarrheal infants fell to the bottom of their non-diarrheal peers. The differences between diarrheal and non-diarrheal infants remained apparent at the 9-month and 12-month time points as well. The peak difference in dorsal lengths between the vaccinated and

unvaccinated infant groups was observed at 9 months of age with a 2.7% improvement among the Vaccinated cohort, resulting in an average 1.3 cm increase in size. By 12 months of age, there was evidence of catch-up growth resulting in a smaller 0.7% difference in linear growth and these observations appear similar to the catch-up growth patterns observed among human children with a history of infant growth stunting[33]. Histograms provide a helpful approach to visualizing general growth rate patterns between groups[34], and Fig. 5b shows the overlay of vaccinated and unvaccinated infant dorsal lengths at 1 month and 9 months of age, representing the baseline and peak difference in dorsal length measurements.

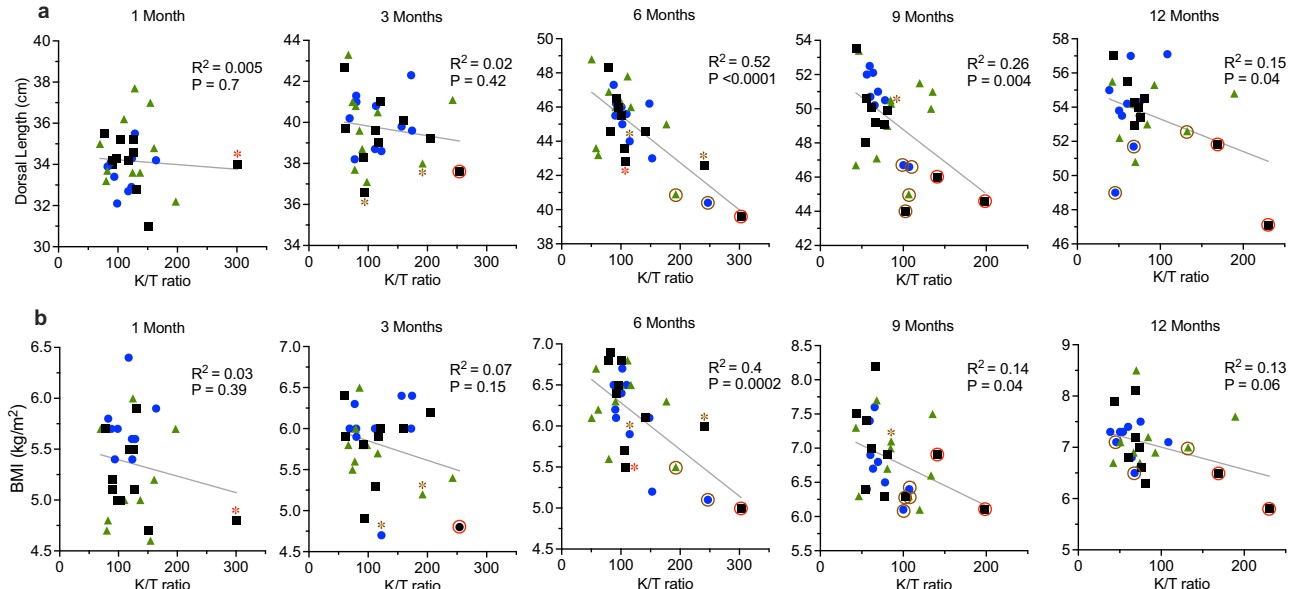

**Fig. 6 | Association between dorsal length and BMI with serum K/T ratios.** Infant dorsal length (**a**) and body mass index (BMI) (**b**) was compared to serum kynurenine/tryptophan (K/T) ratios among a subset of Vaccinated infants and unvaccinated Controls. Symbols represent the same group designations as shown in Fig. 5 and those with an asterisk (*) indicate animals that appeared healthy at the time of measurement but were hospitalized with diarrhea at the next following time point. Brown asterisks indicate future hospitalization with all-cause diarrhea and red asterisks indicate future hospitalization with *C. coli* diarrhea. The study began with 10 vaccinated infants/group (*n* = 10 VDI-1/3/12 infants with 9 remaining at both 9 and 12 months and *n* = 10 VI-1/3/12 infants with 9 remaining at 12 months) and 10 unvaccinated Controls (*n* = 10 Control infants with 9 remaining at 12 months). Brown circles indicate animals that had been hospitalized with all-cause diarrhea whereas red circles indicate animals that were hospitalized with *C. coli*-associated diarrhea. *P* values were determined by univariable linear regression. Source data are provided as a Source Data file.

Although dorsal length measurements represent the gold standard for measuring infant growth stunting, weight measurements were also taken at each time point and there was a significant correlation between weight and length measurements (Supplementary Fig. 4a, *P* ≤ 0.02). At 1 month of age, infants born to vaccinated dams (VDI-1/3/12) had a significantly higher BMI compared to infants born to unvaccinated dams in the Control group (ANOVA, post-hoc two-sided Tukey adjusted, *P* = 0.05) or the VI-1/3/12 group (ANOVA, post-hoc two-sided Tukey adjusted, *P* = 0.03) (Supplementary Fig. 4b). Interestingly, there was no correlation between *Campylobacter*-specific IgA levels in milk and corresponding infant BMI at 1 month of age (Supplementary Fig. 4c, *P* = 0.14) but there was a significant correlation between infant BMI and *Campylobacter*-specific serum IgG levels at this time point (Supplementary Fig. 4d, *P* = 0.01).

Direct comparisons of raw linear growth values (i.e., cm of length) at the various time points were not statistically significant between groups (*P* ≥ 0.19, student's *t* test) but these raw values do not represent the standard approach for measuring human infant growth rates [e.g., length-to-age Z-scores (LAZ)] and lack proper adjustment for confounding variables such as hospitalization with non-*C. coli* diarrhea, initial length at 1 month of age, or sex, since males grow faster than females[24]. We, therefore, analyzed linear growth patterns of the infant macaques in terms of LAZ scores to match the approach used for measuring human infant growth trajectories[34–36]. Since there was no published LAZ database for infant macaques, we prepared this data based on the unvaccinated controls that were measured at the 1, 3, 6, 9, and 12-month time points and adjusted for LAZ at 1 month of age, sex, and non-*C. coli*-associated diarrhea. We observed that by 9 months of age, infants born to vaccinated dams (VDI-1/3/12) had a 1.28 LAZ score improvement compared to unvaccinated controls that was significant (Tukey adjusted, *P* = 0.001). Vaccination of infants alone (VI-1/3/12) did not provide an improvement in growth kinetics (Tukey adjusted, *P* = 0.70). However, at 12 months of age, significant improvements in LAZ scores were still observed between VDI-1/3/12 vaccinated infants

vs. unvaccinated controls (Tukey adjusted, *P* = 0.025). This indicates that among infant macaques that are colonized with *Campylobacter* early in life (Table 1), vaccination of both the pregnant dam and their infant together provided the highest initial infant BMI and later afforded the best antibacterial protection associated with improved growth trajectories.

Growth stunted children in Tanzania[37], Bangladesh[38,39], and Brazil[40] were found to have higher serum kynurenine/tryptophan (K/T) ratios compared to their non-stunted peers and this represents one of the few biomarkers indicative of future growth trajectories[41]. We had also previously noted a significant increase in K/T ratios among growth faltering infant macaques[20] and for these reasons, we measured K/T ratios among the infants monitored for dorsal length and body mass index (BMI) (Fig. 6). At 1 month of age, there was no significant association between dorsal length and K/T ratio (*P* = 0.70, Fig. 6a) or between BMI and K/T ratio (*P* = 0.39, Fig. 6b). Interestingly, the unvaccinated 1-month-old Control infant that would later be hospitalized for diarrhea at 79 days of age (red asterisk) was of average length (34.0 cm, Fig. 6a) but already had a high K/T ratio (K/T ratio = 301) and was among the leanest animals in terms of BMI (Fig. 6b). At 3 months of age, there was still no correlation between dorsal length or BMI of infant macaques with their K/T ratios (*P* = 0.42). However, the Control infant that had recovered from *C. coli* diarrhea (red circle) was among the smallest animals in terms of either dorsal length or BMI. Likewise, two more infants that would later be hospitalized with non-*C. coli* diarrhea before 6 months of age (brown asterisks) also began to trend near the bottom of their peer groups in terms of both dorsal length and BMI. Infant RM are typically weaned between 6 and 9 months of age and at the 6 month time point, there was a highly significant inverse correlation between dorsal length and K/T ratios (*P* < 0.0001) and this correlation was also observed at 9 and 12 months of age (*P* = 0.004 and *P* = 0.04, respectively, Fig. 6a). Likewise, significant inverse correlations were also found between BMI and K/T ratios at 6 and 9 months of age (*P* = 0.0002 and *P* = 0.04, respectively) and this

trended towards significance at the 12 month time point ($P = 0.06$, Fig. 6b). An association between circulating tryptophan levels and future linear growth velocity among human infants has also been noted[37] and likewise, we found a significant 30% increase in serum tryptophan concentrations among VDI-1/3/12 infants compared to Control infants at the 6 month time point (44.3 μM vs. 34.0 μM, respectively, $P = 0.02$, Supplementary Fig. 5), which preceded the 9 month time point that showed the greatest difference in linear growth patterns (Fig. 5). In contrast, there was no significant difference between the tryptophan levels of VI-1/3/12 infants vs. Control infants at any time point ($P \geq 0.58$, Supplementary Fig. 5).

## Discussion

Globally, diarrheal diseases cause extensive morbidity and mortality with a disproportionate impact observed among infants and small children living under conditions of poor sanitation and hygiene. *Campylobacter* spp. represent a leading cause of childhood diarrhea in addition to an association with infant growth stunting. In these studies, we examined the potential role of *Campylobacter* vaccination in reducing diarrheal disease among outdoor-housed infant RM. We found that *Campylobacter* vaccination elicited higher serum antibody responses than those observed after natural infection and vaccination significantly reduced the rates of *Campylobacter*-associated diarrhea and all-cause diarrhea-associated mortality. Moreover, *Campylobacter* vaccination significantly improved infant macaque growth trajectories, demonstrating that vaccination against this single enteric pathogen may not only decrease overt diarrheal disease, but also reduce infant growth stunting.

RM serve as an important model for studying acute and chronic enteric human disease. In addition to being (1) omnivores with similar genetics to humans, these animals are (2) endemic for major human enteric diseases associated with growth stunting including *Campylobacter*, *Shigella*, *Cryptosporidium*, and *Giardia*[20], (3) show histological evidence consistent with EED at an early age[20], (4) demonstrate increased serum K/T (kynurenine/tryptophan) ratios among growth faltering infants[20], and (5) have dystopic microbiome profiles that are similar to those observed in low- and middle-income countries (LMIC) among those who live under conditions of poor sanitation and hygiene[17–19]. In Bangladesh, one study found that over 70% of children between 18 and 24 months of age are colonized with *Campylobacter*[15]. In comparison, RM represent a particularly robust model for *Campylobacter*-associated disease[27] with 76–86% of infant macaques colonized with *Campylobacter* by 1 month of age and 96% of the animals colonized by 3 months of age (Table 1). In our studies, 20–24% of outdoor-housed infant macaques were hospitalized for *Campylobacter*-associated diarrhea (Fig. 3 and Supplementary Fig. 3) and this resulted in a nearly 3% mortality rate by 18 months of age (Fig. 4) despite routine monitoring, rapid diagnosis/treatment, and excellent veterinary care. *Campylobacter* vaccination of pregnant dams and their offspring reduced *Campylobacter*-associated diarrheal disease by up to 79% in the per-protocol population that approached statistical significance (Fig. 3, $P = 0.09$) with a significant overall vaccine efficacy across all vaccination groups reaching 55% ($P = 0.02$). These numbers are similar to the results observed in our prior study in which entire rhesus macaque shelter groups (all age groups; only 18% infants) were vaccinated against *Campylobacter* and vaccine efficacy reached 83% ($P = 0.048$)[27]. The *Campylobacter* vaccine used in these current infant vaccination studies utilized the peroxide-based HydroVax inactivation approach in 2017 whereas in 2018, the bacteria were inactivated using the HydroVax approach in the presence of formaldehyde as a stabilizing agent[42]. Based on the ELISA titers observed at 2 months after primary vaccination (i.e., 3 months of age) in 2017 vs. 2018, there was no significant difference in *Campylobacter*-specific antibody levels (37,500 EU vs. 38,200 EU, $P = 0.50$, student's *t* test Fig. 2a). More studies are underway to determine if changes in VE observed between

these studies were due to differences the age/susceptibility of the study population (infants vs. mainly juveniles/adults) or to differences in circulating *Campylobacter* strains.

This study has several limitations. For example, the vaccine strain of *C. coli* (NTICC13, isolated in 2013) was a mismatch for the LOS and CPS of the circulating strains of *C. coli* that would have been encountered during the time of this vaccine study (2017 and 2018)[27] and this may have impacted vaccine efficacy since homologous vaccine-mediated protection may be higher than that observed against heterologous strains of bacteria. Moreover, 76–86% of infant RM are already colonized/infected with *C. coli* prior to their first vaccination at 1 month of age and it is possible that if they could have been vaccinated earlier, or if initial infection with *C. coli* occurred later in infancy, then higher VE may have been established. It would have been interesting to measure growth kinetics of *Campylobacter*-negative animals to *Campylobacter*-positive animals, but these types of studies were not possible since 100% of the infant macaques were colonized by 12 months of age (96% of infants were colonized with *Campylobacter* by 3 months of age). An immunological correlate of immunity has not been identified, but determining the potential role for B cell-mediated immunity vs. T cell-mediated immunity may be best determined by antibody-mediated depletion studies[43] or passive transfer of immune serum[43,44] in an optimized direct oral challenge model rather than in natural repeat-exposure field studies like those performed here, since these address different experimental questions. Another limitation was small sample size. Unlike direct-challenge models that, once optimized, can be performed with relatively few animals, field studies involving natural exposure/infection often require larger cohorts but have the potential advantage of more closely mirroring human exposure/infection under conditions of poor sanitation and hygiene. In this natural exposure field study, there was a trend towards high vaccine-mediated protection against hospitalization with *C. coli* diarrhea in the VDI-1/3/12 group (Per-protocol VE = 79%, $P = 0.09$) and when all vaccinated infant groups were combined, Per-protocol VE reached 55% ($P = 0.02$, Fig. 3). In these studies, we did not measure *Campylobacter*-specific serum IgA but instead focused on measuring serum IgG levels since serum IgG is more likely to be elicited following intramuscular vaccination and more likely to be involved in protection against enteric pathogens in this model. For example, the VDI-1/3/12 vaccine group showed the highest trend in protection against *C. coli* diarrhea (Fig. 3) and was responsible for statistically significant improvements in infant growth stunting (Fig. 5). This may be due, at least in part, to the 7-fold higher levels of serum IgG among infants born to vaccinated dams (4560 EU for VDI-1/3/12 infants) at 1 month of age compared to infants born to unvaccinated dams (628 EU for either VI-1/3/12 or VI-1/3/5 infants, respectively) (Fig. 2) since the dams were vaccinated during pregnancy and serum IgG (but not serum IgA) is transferred across the placenta to provide early protection to the newborn[45]. Vaccine-induced serum IgG may also be important and it is possible for serum IgG to transudate to the gut and provide protection against enteric pathogens. For example, anti-rotavirus serum IgG administered by passive transfer/intravenous administration provided protection to pigtailed macaques following subsequent oral rotavirus infection[44] and studies in mice have shown that the majority of intestinal IgG is serum transudate or bile-derived whereas ≥98% of intestinal IgA is of the secretory type and is not serum-derived[46]. It is unclear if *Campylobacter*-specific IgA in milk from vaccinated dams represented a mechanism of protection associated with the improvements in infant growth trajectories in the VDI-1/3/12 group since anti-bacterial IgA titers from the vaccinated dams was not significantly higher than that observed among unvaccinated dams in the VI-1/3/12 or Control groups (Supplementary Fig. 2) and was not associated with increased infant BMI at 1 month of age (Supplementary Fig. 4c). This is in contrast to *Campylobacter*-specific serum IgG levels that did correlate with an improved infant BMI at 1 month of age (Supplementary

Fig. 4d). However, antibacterial IgA levels in RM milk may have been higher at earlier time points since clinical studies have shown that antibacterial IgA levels can be several fold higher in early colostrum compared to later milk samples from mothers immunized with Tdap (tetanus, diphtheria, acellular pertussis) or meningococcal (*Neisseria meningitidis*) vaccines during pregnancy[47,48] and more studies are needed to assess these early time points in relation to the positive infant outcomes observed in this rhesus macaque model.

Over 500,000 children under 5 years of age die each year due to diarrheal disease[1] and *Campylobacter* itself is directly responsible for as many as 37,000 deaths per year[2]. In these studies, we monitored diarrhea-associated mortality rates among >300 vaccinated and unvaccinated infant macaques for up to 18 months and across two birth cohorts (2017 and 2018). There were 7 *C. coli*-associated deaths among 248 unvaccinated animals compared to no *C. coli*-associated deaths among 90 *Campylobacter*-vaccinated infants, but this did not reach statistical significance (100% VE, $P = 0.2$). Although this is a promising preliminary result, more studies that are statistically powered to address this question are needed. Infant macaques are exposed to a number of enteric pathogens and in terms of all-cause diarrhea-associated mortality, there were 23 diarrhea-associated deaths among the unvaccinated animals (23/248, 9.3%) compared to only 2 diarrhea-associated deaths among the *Campylobacter*-vaccinated infants (2/90, 2.2%), resulting in 76% VE (Fig. 4, $P = 0.03$). Although the underlying mechanisms for these results remain unclear, it seems unlikely that vaccination against *Campylobacter* provides direct cross-reactive immunity against other unrelated enteric microbes (e.g., *Shigella*, *Giardia*, *Cryptosporidium*, etc.). However, it is possible that vaccination against a common enteric bacterium such as *Campylobacter* may result in reduced gut pathology/inflammation and this, in turn, may reduce the risk of severe invasive disease by other enteric pathogens. This hypothesis is not unique to mucosal pathogens like *Campylobacter*; for instance, susceptibility to HIV is increased under conditions of non-specific inflammation caused by bacterial vaginitis[49] or by co-infections with other STDs[50] and so it is plausible that a similar mechanism may play a role in this experimental model as well. Nevertheless, more studies are needed to better understand the potential off-target benefits of *Campylobacter* vaccination, and we believe that this represents an area worthy of further study.

The options for treating infant growth stunting have been limited, especially since a number of nutritional and public health interventions have failed to improve linear growth rates of infants in LMIC. In a network meta-analysis of 29 randomized clinical trials (RCT) of infants aged 0–6 months[51], interventions ranging from administration of micronutrients, food supplements, deworming agents, increasing water sanitation and hygiene (WASH), or improving maternal education were compared. Of these interventions, an improvement of 0.2 LAZ was identified after administration of multiple micronutrients whereas the other interventions did not facilitate a significant improvement in infant growth kinetics compared to standard of care. In a similar network meta-analysis of 79 RCT among infants aged 6–24 months[52], multiple micronutrients provided an improvement of 0.06 to 0.08 LAZ whereas food supplements including flour or fortified lipid-based nutrient supplements did not improve LAZ. Likewise, deworming agents, initiating WASH protocols, or improving maternal education did not provide an improvement in LAZ. Daily egg-based nutritional supplementation of 6–9 month-old Ecuadorian infants showed initial promise with a sizeable improvement of 0.63 LAZ[34], but the improvements in linear growth were no longer present after 2 years[53], and similar nutritional studies performed in Malawi failed to show an improvement in linear growth kinetics[54]. Other sophisticated approaches to nutritional intervention such as the development of highly defined microbiota-directed complementary food (MDCF) prototypes have been shown to increase expression of certain biomarkers associated with growth, but two clinical trials failed to show an

improvement in LAZ score over standard of care nutritional supplementation[35,36]. A small improvement in weight-for-length changes in Z scores was observed in one study[36], but some have questioned the implications of this advanced approach by noting that in addition to this dietary intervention having no significant improvement in height gain, "The difference in weight gain has little clinical significance: an additional 2 years of MDCF-2 supplementation would still be needed to shift the weight-for-length distribution toward the normal range."[55] This raises an important point: once growth stunting has occurred, it is difficult to substantially accelerate an infant's growth kinetics to catch up to their non-stunted peers. In contrast, preventative interventions such as successful vaccination against enteric pathogens associated with growth stunting, may provide increased clinical benefit by preventing growth stunting before it starts.

What is the underlying cause for poor growth trajectories and infant growth stunting? It appears to not simply be a question of providing a more nutritious diet since several clinical trials of various nutritional interventions have failed to improve growth stunting[34–36,51–54]. This would suggest that malabsorption of nutrients is a major driver of poor infant growth. Among healthy individuals, the small intestine is the main site of nutrient absorption and children in LMIC often have gut pathology including villous blunting, villus atrophy and reduced crypt-to-villus ratios that would be anticipated to reflect poor nutrient uptake by the small intestine. These observations would seem to explain poor nutritional uptake among growth stunted infants, but this could also be an epiphenomenon wherein histopathology of the small intestine may not be the sole factor involved with nutrient absorption and a number of studies have drawn this into question. For instance, a post-mortem histological analysis of the small intestine among children living in India in 1969 found that although the GI tract of stillborn infants was essentially pristine, the small intestine of neonates and children showed substantially more histological abnormalities in the duodenum and ileum[21]. However, it was noted that the two children in their study that suffered from protein-calorie malnutrition had histopathology/mucosal architecture that was no different from the other children that died of other causes[21]. More recent histological analyses of duodenal biopsies of children in Pakistan and Zambia were performed to determine if there was an association between small intestine histopathology and clinical severity[56], but no significant association between duodenal biopsy scores and growth stunting was found. Similar results were observed among outdoor-housed RM in which 1-day-old infant macaques had pristine small intestinal architecture that was in stark contrast to 8–11 month old infants in which nearly all of the animals demonstrated the classic EED-associated histological abnormalities observed among afflicted human infants including villous blunting, villus atrophy and reduced crypt-to-villus ratios[20]. In the macaque model, the histological architecture of the entire GI tract was measured and similar to the Liu et al. study that quantitatively examined histopathology of duodenal biopsies[56], we likewise found no correlation between growth faltering among infant macaques and histological abnormalities in the duodenum, jejunum, ileum (proximal, mid, and distal) or when the entire small intestine score was calculated ($P \geq 0.28$). In contrast, there were significant correlations between histological abnormalities in the large intestine and reduced infant macaque growth kinetics including the cecum ($P = 0.02$), ascending colon ($P = 0.02$), transverse colon ($P = 0.03$), descending colon ($P = 0.07$; trending towards significance), rectum ($P = 0.004$), and the large intestine in total ($P = 0.01$)[20]. Several clinical studies of children living under conditions of poor sanitation and hygiene have found an association between high serum K/T ratios and infant growth stunting[37–41]. This is similar to the results observed in our studies involving the growth kinetics of outdoor-housed infant RM in which there is a significant inverse correlation between K/T ratios and dorsal length (Fig. 6a) or BMI (Fig. 6b). In our previous histological analysis of the full-length infant rhesus macaque GI tract, we found no

correlation between serum K/T ratios and histopathology at any site along the small intestine ($P \geq 0.26$) whereas high K/T ratios were correlated with histopathology in the large intestine including the transverse colon ($P = 0.001$, $R^2 = 0.67$), descending colon ($P = 0.001$, $R^2 = 0.69$), rectum ($P = 0.01$, $R^2 = 0.47$), and the large intestine in total ($P = 0.01$, $R^2 = 0.47$)[20]. Together, this indicates that infant growth stunting is associated with high K/T ratios, and high K/T ratios are associated with histological abnormalities in the large intestine. This would suggest that infant growth stunting is not strictly a disease of the small intestine and may instead be associated with damage to the large intestine. If these results are confirmed, and future studies indicate a likewise significant reduction in large intestine pathology due to *Campylobacter* vaccination, then this would represent a paradigm shift in our current understanding of infant growth stunting under environmental conditions of poor sanitation and hygiene. Clearly more studies are needed, but it is interesting to note that in other diseases such as pediatric short bowel syndrome in which nutrient absorption by the small intestine is compromised due to surgical resection, the colon/large intestine becomes critical for energy salvage[22,57,58].

The high incidence of *Campylobacter* colonization and diarrheal disease among RM in the presence of endemic exposure to a number of other human enteric pathogens (e.g., *Shigella*, *Cryptosporidium*, *Giardia*) provides a robust model for measuring interventions aimed at reducing enteric disease and infant growth stunting under conditions that mimic examples of poor sanitation and hygiene. Another advantage of the rhesus macaque model is that these animals develop and mature at approximately 3–4-fold faster rates than humans, reaching reproductive maturity by ~4 years of age with an average lifespan of approximately 25 years when raised in captivity[59]. Based on a 4-fold faster maturation rate compared to humans, this indicates that a 9-month old rhesus macaque would be developmentally similar to a 2.5 year old child; a 1-year-old macaque would be similar to a 4-year-old child; and monitoring diarrheal mortality for up to 18 months of follow-up would be similar to following children to 6 years of age. Interestingly, we found the greatest vaccine-associated improvement in linear growth stunting at 9 months of age (2.7% difference, Tukey adjusted, $P = 0.001$) and although the differences were smaller by 12 months of age, they were still statistically significant (0.7% difference, Tukey adjusted, $P = 0.025$). Similar observations of catch-up growth among growth stunted children in a MAL-ED cohort study have also been identified by 5 years of age[33]. Indeed, among the children who were stunted at 24 months ($n = 426$), 185 (43%) were no longer stunted at 60 months. This suggests that the reduced difference infant macaque growth stunting observed at a similar stage of pediatric development (i.e., 1 year of age for macaques vs. 4–5 years of age for humans) appears to be mimicking a similar pattern of early linear growth stunting followed by later catch-up growth. Since diarrhea-associated mortality rates are highest among children <5 years of age, monitoring infant macaques for 12–18 months provides a more rapid assessment of experimental intervention success (or failure) in an expedited time frame in addition to providing an opportunity to measure gut histology/inflammation in more detail than can be performed during most clinical studies[20]. Together, the results provided here indicate that *Campylobacter* vaccination of infant RM not only provides protection against diarrhea-associated morbidity and mortality but also reduces infant growth stunting by improving dorsal length growth kinetics.

## Methods
### Rhesus macaques
All animal work was approved by the ONPRC Institutional Animal Care and Use Committee (IACUC protocol: IP00000416) and performed in strict accordance with the recommendations described in the Guide for the Care and Use of Laboratory Animals of the National Institute of Health, the Office of Animal Welfare and the United States Department of Agriculture. RM (*Macacca mulatta*) were housed in outdoor sheltered breeding groups consisting of approximately 20–50 animals per shelter[23,24]. Animals were fed twice daily with a standard commercial primate chow, water was available *ad libitum* and the sealed concrete floors of the group enclosures were washed daily. Infants were observed daily for disease or injury by husbandry technicians including staff who were not associated with the study and cases were brought to veterinary staff for evaluation. Infants with liquid stool were hospitalized if they had further signs of illness including: dehydration as evidenced by lethargy, weak grip on dam, not holding on with all limbs, sunken dull eyes, slow blinking or dragging eyelids, lack of alertness or interest in surroundings, more interest in sleeping than nursing, or rough hair coat appearance[24]. Infants born in outdoor shelters from March through July in 2017 and 2018 were included in the analysis with the following exclusions: No animals were used from shelter groups that were involved with prior *Campylobacter* vaccination studies[27] or receiving experimental probiotic treatments for alopecia. Animals were monitored for diarrhea-associated hospitalizations and mortality for up to 18 months after birth using electronic health records and were considered lost to follow-up if they were housed indoors for >50% of their lifespan, died, required euthanasia for unrelated reasons, or were transferred to other research studies. Time-to-event studies were performed for diarrheal incidence ($n = 103$ vaccinated infants, $n = 266$ unvaccinated control infants) whereas diarrhea mortality studies were performed only with animals that were not euthanized due to a scheduled protocol-directed necropsy ($n = 90$ vaccinated infants, $n = 248$ unvaccinated control infants). Adult females (dams, $n = 61$) were also included for measuring microbiome and antibacterial immune responses. Animal age range: 1 month to 18 years old. Infant RM are prone to chronic and relapsing diarrhea, making it difficult to reliably diagnose the causative agent during a secondary or tertiary case of diarrhea because it could be a relapse related to a prior case. For this reason, infants were either scored as *C. coli*-positive/all-cause diarrhea-positive or *C. coli*-negative/all-cause diarrhea-positive at the time of first hospitalization and this represented the endpoint of their diarrhea-free days in the Kaplan Meier curve analysis.

A subset of vaccinated and unvaccinated animals was monitored longitudinally for dorsal length growth kinetics, weight gain kinetics, and diarrheal episodes with rectal swabs screened for *C. coli, C. jejuni, Shigella*, and *Salmonella* by microbial culture on differential medium. These infants were chosen based on their health (no major injuries or broken bones at 1 month of age) and the hierarchy of their dams as well as their infant-rearing history in consultation with ONPRC animal husbandry staff who were not involved in the study and did not know whether the animals would be in a vaccination group or a control group. Although the longitudinal infants were not formally randomized, they were typically chosen to be included in a vaccine or control group prior to their first physical exam and in all cases their length and weight were not known until after they were enrolled within a particular study group. Dorsal length measurements were performed using a Seca 210 measuring mat developed for measuring human infants and toddlers (Seca, product number 210 1721 004). Following ketamine or Telazol anesthesia of the infant, two individuals work together to perform the measurements. The infant was placed on the measuring mat on its back with its head in the center of the head positioner. The legs were then fully extended with the toes pointed up before locking the foot positioner in place and reading the dorsal length in centimeters. Although the measurers were not blinded to the treatment group, they were unaware of the prior length measurement of an animal at the time of each new longitudinal measurement. All procedures, including dorsal length measurements, weight

measurements, vaccinations, blood draws and rectal swabs were performed under ketamine or Telazol anesthesia by trained personnel under the supervision of veterinary staff. BMI was defined as kilogram/meter$^2$ and was calculated using the following online calculator: https://www.nhlbi.nih.gov/health/educational/lose_wt/BMI/bmi-m.htm.

## Campylobacter vaccination

*C. coli* represents the predominant *Campylobacter* species at the ONPRC and an inactivated *Campylobacter* vaccine was prepared from a 2013 isolate (NTICC13) grown under microaerophilic conditions to late-log phase in shaker flasks and concentrated/purified by tangential flow filtration. In 2017, the *Campylobacter* vaccine was prepared by inactivating the bacteria with 0.01% $H_2O_2$, 2 μM $CuCl_2$, 20 μM methisazone in a PBS buffer containing 150 mM $NaPO_4$. Prior studies involving pertussis toxin indicated that brief exposure to formaldehyde could improve protein stability and immunogenicity[42] and in 2018, the inactivation approach was modified to include 0.005% $H_2O_2$, 2 μM $CuCl_2$, 20 μM methisazone and 0.04% formaldehyde in a PBS buffer containing 150 mM $NaPO_4$. In both cases, inactivation was performed for 20–22 h at room temperature and a 40 μg dose was formulated with 0.1% alum[27]. The immunogenicity of vaccine lots prepared in 2017 and 2018 appeared to be similar since *Campylobacter*-specific IgG titers among vaccinated infants in each birth cohort were not significantly different at 3 months of age [i.e., 2 months post-primary vaccination (Fig. 2a); geometric mean: 37,400 EU versus 38,200 EU, in 2017 vs. 2018, respectively, *P* = 0.50, student's *t* test]. Similar results were also observed when measuring *C. coli* flagellin-specific responses at 2 months after primary vaccination (Supplementary Fig. 1a; geometric mean: 15,600 EU vs. 14,700 EU, in 2017 vs. 2018, respectively, *P* = 0.85, student's *t* test). In 2017, 30 dams from the VDI-1/3/12 group were vaccinated twice by intramuscular administration, 28 days apart during pregnancy. Three dams did not have offspring, due to not being pregnant (i.e., palpation negative at the time of vaccination) or pregnancy loss. One dam gave birth pre-term and one infant birth was not recorded in time for vaccination, resulting in exclusion from the study. These results are in agreement with expected fecundity rates among RM at the ONPRC. Groups of infants were immunized intramuscularly with a 40 μg dose of *Campylobacter* vaccine: VDI-1/3/12 (*n* = 25, 52% female), VI-1/3/12 (*n* = 28, 54% female), and VI-1/3/5 (*n* = 50, 56% female). An additional 25 contemporaneous unvaccinated Controls (*n* = 25, 44% female) were monitored closely in parallel. An average of 3 infants (range, 1–15) were vaccinated against *Campylobacter* per shelter group in 2017 and an average of 2 infants (range, 1–5) were vaccinated against *Campylobacter* per shelter group in 2018.

## Serum tryptophan and kynurenine measurements

Serum tryptophan and kynurenine levels were measured in a blinded manner by technical staff at the ONPRC Endocrine Technologies Support Core (ETSC) using ultra-high performance liquid chromatography-heated electrospray ionization-tandem triple quadrupole mass spectrometry (LC-MS/MS) on a Shimadzu Nexera-LCMS-8050 instrument (Kyoto, Japan)[20]. For sample preparation, 10 μl of sample was combined with 10 μl of internal standard mixture (10 μg/ml trpytophan-d3 in 0.1% formic acid) and extracted with 100 μl of cold methanol for 30 min on ice. Following extraction, samples were centrifuged, filtered, dried under forced air, reconstituted in 100 ul of 5:95 methanol:water with 0.1% formic acid and placed onto 96-well microtiter plates. A quality control pool of rhesus macaque serum was prepared in the same manner as samples and assayed in quadruplicate on each plate. A 6-point standard curve from 40 μg/ml to 0.0128 μg/ml was prepared in charcoal-stripped human serum (Biochemed Services, Winchester, VA) demonstrated to be free of tryptophan and kynurenine. Standard curve points were extracted and assayed in triplicate in the same manner as the samples. A quality

control sample containing 25 ng/ml of each standard in 5:95 methanol:water with 0.1% formic acid was run daily before and after each assay to confirm system suitability.

After the reconstitution step, samples were subjected to LC-MS/MS analysis. Using a Shimadzu SIL-30CAMP autosampler set to 10 ˚C, 5 μl of sample was injected onto an Ace Excel 2 C18-PFP column (50 × 2.1 mm). Chromatographic separation occurred at 15 ˚C at a flow rate of 500 μl/min. Using a Shimadzu Nexera LC30-AD system, a linear gradient starting at 5% mobile phase B and ending at 95% mobile phase B was run for 2.25 min after a 0.30-min hold at 5% B. Mobile phase A (0.1% formic acid, pH 2.6) and mobile phase B (0.025% formic acid, 5 mM ammonium formulated in methanol) were chosen after optimization to minimize carryover while also maximizing detector signal and column retention. Heated electrospray ionization (ESI) interface settings were optimized for signal and stability. The interface temperature was 300 °C, desolvation line temperature was 200 °C, and heat block temperature was 500 °C. Gas was supplied by a Peak Genius 1051 nitrogen and air generator. Nitrogen gas was used for nebulizing and drying gases, while air was used for heating gas. Nebulizing gas flow was set at 2 L/min, heating gas flow was set at 10 L/min, and drying gas flow was set at 10 L/min. Interface voltage was set to 1 kV. Scheduled multiple reaction monitoring (MRM) transitions were collected using a Shimadzu LCMS-8050 tandem triple quadrupole MS in positive mode with two MS transitions for each analyte at their respective retention times: tryptophan (205.00 > 146.10, 205.00 > 118.20) at 1.98 min, kynurenine (209.10 > 94.15, 209.10 > 146.05) at 1.49 min, and trptophan-d3 (208.15 > 147.00, 208.15 > 119.15) at 1.98 min. Data processing and analysis was performed using LabSolutions Software, V5.72 (Shimadzu, Kyoto, Japan). Extraction recovery was 80% for both amino acids. Intra-assay variation based on the rhesus macaque QC pool was less than 5% and inter-assay variation was less than 6% for both tryptophan and kynurenine.

## Enzyme-linked immunosorbent assay (ELISA)

ELISA plates were coated with an optimized concentration of HydroVax-inactivated *Campylobacter* whole-cell lysate (1 μg/mL) or semi-purified flagellin antigen (1 μg/mL)[27] except that serum samples were not preadsorbed with *Shigella* prior to performing the assay since this did not appear to alter assay performance. The concentration of each *Campylobacter* antigen was optimized in small-scale pilot studies using serial dilutions of protein to provide the highest sensitivity with the lowest background. The *Campylobacter* whole-cell lysate was prepared from late log-phase *C. coli* (strain: NTICC13) grown in suspension overnight and homogenized in PBS (Fisher PowerGen 125 homogenator, setting 5 with homogenization performed 3 × 60 s/each with cooling on ice in between) to prepare a bacterial lysate that was aliquoted and stored at −80 °C. To prepare semi-purified flagellin antigen, an approach was adapted from in ref. 60 in which the homogenized lysate was centrifuged twice at 2000 × *g* for 30 min, with the pellet resuspended in PBS each time. The pooled supernatants were then processed by ultracentrifugation for 3 h at 100,000 × *g*. Semi-purified flagellin pellets were resuspended in PBS, aliquoted, and stored at −80 °C. Once the optimized concentration of antigen was determined, ELISA plates were coated overnight at 2–8 °C in bulk (i.e., 20–80 ELISA plates/batch) and then stored at −20 °C until use. Plates were thawed and unbound antigen was removed before plates were blocked with 5% non-fat dry milk in PBS-T (PBS supplemented with 0.05% Tween-20) for 1 h at room temperature. Plates were rinsed 1X with PBS-T and incubated for 1 h with 3-fold serial dilutions of RM breast milk or heat-inactivated serum. Fifty microliters of 10% $H_2O_2$ was added (3% $H_2O_2$, final concentration) to each well and incubated for 30 min at room temperature to inactivate potential blood-borne pathogens. Plates were washed 3X with PBS-T and incubated with an optimized dilution (1:2000) of a mouse anti-rhesus macaque IgG-HRP antibody (clone SB108a, Southern Biotech, Cat. No. 4700-05) or an

optimized dilution of 1:1000 of a goat anti-rhesus macaque IgA-HRP antibody (Cat. No. orb21435, Biorbyt) for 1 h at room temperature. Plates were rinsed 3X with PBS-T and 100 µl of o-phenylenediamine dihydrochloride (OPD, 0.04%) substrate containing 0.01% $H_2O_2$ in citrate buffer (0.05 M citric acid, 0.1 M $Na_2HPO_4$, pH = 5.0) was added for 20 min, with colorimetric development stopped by the addition of an equal volume of 1 M HCl. Optical densities (OD) were measured at 490 nm using a VersaMax ELISA plate reader (Molecular Devices) with SoftMax Pro software V6.5.1 and a log–log transformation of the linear range of OD (0.05 to 1.5 OD) versus reciprocal serum dilution was performed and end-point ELISA titers were determined as the reciprocal of the serum dilution needed to reach an OD = 0.10 after background subtraction. Each ELISA plate included a *Campylobacter*-immune serum standard that was serially 3-fold diluted in duplicate on each plate to allow normalization between ELISA plates performed in the same experiment or between experiments performed on different days or with ELISA plates coated with different production lots of antigen. Each experimental serum sample was tested at least in duplicate and paired samples with >25% coefficient of variation (CV) were repeated. The operator was blinded in terms of group designation at the time they were performing the assay. The *Campylobacter*-immune serum IgG and IgA serum standard was from a rhesus macaque obtained at 5.7 years after recovery from 6 prior *C. coli*-associated hospitalizations for diarrhea.

## 16 S amplicon sequencing

Total DNA was extracted from rectal swabs using the DNeasy Powersoil Pro Kit (Qiagen, Valencia, CA, USA) and included 7–37 samples per group for each time point (1, 3, 6, and 9 months). The hypervariable V4-V5 region of the 16 S rRNA gene was amplified using PCR primers (forward primer, 515 F: GTGYCAGCMGCCGCGGTAA and reverse primer, 926 R: CCGYCAATTYMTTTRAGTTT with the forward primers including a 12-bp barcode) purchased from Integrated DNA Technologies (Coralville, Iowa, USA). PCR reactions were conducted in duplicate and contained 12.5 µl GoTaq master mix, 9.5 µl nuclease-free $H_2O$, 1 µl template DNA, and 1 µl of 10 µM primer mix. Thermal cycling parameters were 94 °C for 5 min, 35 cycles of 94 °C for 20 s, 50 °C for 20 s, 72 °C for 30 s, followed by 72 °C for 5 min. PCR products were purified using a MinElute 96 UF PCR Purification Kit (Qiagen, Valencia, CA, USA). Libraries were sequenced (2 × 300 bases) using an Illumina MiSeq Control Software v3.1.0.

Raw FASTQ 16 S rRNA gene amplicon sequences were uploaded and processed using the QIIME2 analysis pipeline[61]. Briefly, sequences were demultiplexed and the quality filtered using DADA2[62], which filters chimeric sequences and generates an ASV table equivalent to an operational taxonomic unit (OTU) table at 100% sequence similarity. Sequence variants were then aligned using MAFFT[63] and a phylogenetic tree was constructed using FastTree2[64]. Taxonomy was assigned to sequence variants using q2-feature-classifier against the SILVA database (release 138)[65]. To prevent sequencing depth bias samples were rarified to 10,000 sequences per sample before alpha and beta diversity analysis. QIIME 2 was also used to generate the following alpha diversity metrics: richness (as observed ASV). Beta diversity was estimated in QIIME 2 using weighted UniFrac distances[66].

## Microbial cultures

RM had rectal swabs taken at each visit in addition to when infants were hospitalized for diarrhea and these were evaluated in a blinded manner by direct microbial culture on selective media at the ONPRC clinical pathology laboratory which routinely screens for *C. coli, C. jejuni, S. flexneri, S. dysenteriae*, and *Salmonella* spp. If the sample was negative for these specific enteric pathogens, then it is possible that other bacterial, viral or parasitic pathogens may have been present but without formal identification/confirmation, the samples were coded as "normal flora" or "diarrhea of unknown etiology".

## Statistics

To determine whether vaccination altered the overall composition of gut microbiome within any timepoint, we ran PERMANOVAs on Weighted Unifrac distance for 16 S amplicon data using the function ADONIS from the R package Vegan[67]. For single factor comparisons across more than two time points, such as ASV Richness, we analyzed data by fitting a mixed model as implemented in GraphPad Prism 9.3.1. This mixed model uses a compound symmetry covariance matrix and is fit using Restricted Maximum Likelihood (REML). Upon significant main effect, post-hoc tests were conducted using Šídák's multiple comparisons test. The LEfSe algorithm[68] was used to identify differentially abundant taxa between cohorts within timepoint with a logarithmic Linear discriminant analysis (LDA) score cutoff of 2.

For *Campylobacter*-specific antibody titer analysis, mixed effects models were used to compare model-based estimated marginal means of lysate or flagellin (on log scale) in specified contrasts. Pairwise comparisons of antibody titers were based on two-sided student's *t* test with unequal variance and comparisons of 3 or more groups were performed by ANOVA. ELISA values were highly right-skewed and were therefore log transformed prior to determining the linear relationship between IgG or IgA levels in comparison with BMI. Models contain random intercept and fixed effects of combination levels of vaccination group and diarrhea status, time and interaction between these two variables. Cohort effect (2017 and 2018) was not considered in the model because it is partially confounded with vaccination group. Controls from the two cohorts were combined together as control group (no significant differences in diarrheal disease incidence were found between controls from the two cohorts, Supplementary Fig. 3). Post-hoc pair-wise comparisons tailored to specific scientific questions were performed, with Tukey or Dunnett multiple testing adjustment. For *C. coli*-associated diarrhea analysis, log-rank test was used to compare the cumulative risk of *C. coli*-associated disease between the specified vaccinated and unvaccinated groups. VE was defined as (1 − Hazard Ratio [HR]) × 100% where HR was obtained from a Cox PH model on the association between time-to-disease and vaccination group. The 95% confidence intervals for HRs were obtained by inverting the partial-likelihood score test[69]. VE for protection against diarrhea-associated mortality was defined as (1 − Relative Risk [RR]) × 100%, and 95% Taylor series confidence intervals[70] were calculated. *P* values presented were based on Fisher's exact test. For LAZ analysis at 9 months or 12 months of age, the initial multivariable linear model contained the vaccination group (VDI-1/3/12, VI-1/3/12, and control), and possible confounders were examined including sex, LAZ at 1 month of age, and non-*C. coli*-associated diarrhea. Next, an automated backward model selection procedure by AIC was used to select a sensible final model. Group (primary variable of interest) and sex (pre-defined confounder) were retained in the model during the whole process and the final model contains all of the variables listed above. For determining potential associations between dorsal length or BMI with K/T ratios, the coefficient of determination ($R^2$) and *P* values from univariable linear regression were reported. *P* values were two-sided where applicable. Statistical analyses were performed using R4.1.1 (R Foundation for Statistical Computing, Vienna, Austria, 2019), SAS9.4 (SAS Institute, Cary, NC), and GraphPad PRISM (V9.3.1).

## Reporting summary

Further information on research design is available in the Nature Portfolio Reporting Summary linked to this article.

## Data availability

The 16 s rRNA data generated in this study have been deposited in the Sequence Read Archive (SRA) database under accession code PRJNA896946. Source data are provided with this paper.

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

## Acknowledgements

We thank David Erikson and the ONPRC Endocrine Technologies Support Core (ETSC) at the Oregon National Primate Research Center for performing LC-MS/MS assays to measure serum tryptophan and kynurenine levels and Michelle Pounder for performing diagnostic microbial cultures at the ONPRC Clinical Pathology Laboratory. This work was supported, in whole or in part, by the Bill & Melinda Gates Foundation [OPP1149233]. Under the grant conditions of the Foundation, a Creative Commons Attribution 4.0 Generic License has already been assigned to the Author Accepted Manuscript version that might arise from this submission. This work was also supported by US National Institute of Health grant P51 OD011092 for operation of the ONPRC. N.S.R. was supported by NIH T32 AI007319.

## Author contributions

S.M.H. and A.T. were study coordinators, performed data analysis, and prepared data/figures for publication. H.-P.R and J.F.S. performed ELISA assays and data analysis and H.-P.R. prepared figures for publication. K.P., A.J.H., and S.M.H. performed animal health assessments, physical animal measurements, and performed data analysis. N.S.R. and I.M. performed DNA extractions, microbiome assessments, data analysis, and prepared figures for publication. L.G. provided statistical assessment of study parameters. B.K.Q. and I.J.A. cultured Campylobacter, prepared Campylobacter vaccine antigens, provided bacterial antigens for ELISA, and performed data analysis. K.P., A.J.H., L.G., I.M., I.J.A., and M.K.S. were involved with study design and all authors reviewed and edited the manuscript prior to submission.

## Competing interests

OHSU, M.K.S., and I.J.A. have a financial interest in Najít Technologies, Inc., a company that may have a commercial interest in the results of this research and technology. This potential individual and institutional conflict of interest has been reviewed and managed by OHSU and Najít Technologies, Inc. S.M.H., A.T., H.-P.R., K.P., A.J.H., N.S.R., J.F.S., L.G., B.K.Q., and I.M. declare no competing interests. No writing assistance was utilized in the production of this manuscript.
