## [Peer Review File · Nature Communications]

REVIEWER COMMENTS

Reviewer #1 (Remarks to the Author):

This manuscript represents an attempt to show that *Campylobacter* vaccine can reduce diarrhea and growth-stunting in captive juvenile rhesus macaques. This is very timely and human health translatable topic as *Campylobacter*-associated morbidity is significant in both human infants and juvenile (captive) macaques. Before manuscript can be accepted for publication, several specific issues need to be addressed.

Specific Comments:

- 1) Results indicate that vast majority of *Campylobacter* isolates in macaques are linked to *C. coli*. From the past results generated by other groups, there appears to be higher morbidity caused by *C. jejuni* than *C. coli* strains. Did authors try to determine what particular *Campylobacter* strain is exclusively or more associated with clinical disease in macaques? Vaccine appears to be directed broadly against all *Campylobacter*s. More specifically, it would be very informative to identify the components of *Campylobacter* that are functioning as antigenic elicitors of protective immunity.
- 2) Since parasitic and viral pathogens as well as food sensitivities were all identified as potential contributors to clinical diarrheas in juvenile macaques, was there any attempt to determine such %s (even if minor) at ONPRC?
- 3) In results, authors state that vaccination did not affect diversity of gut microbiome. However, it is not clear what type of diversity (alpha or beta) they are referring to. It would be informative to clarify this.
- 4) What was the rationale of focusing on *Campylobacter* serum IgG? Did authors anticipate this metric to correlate with protection against diarrhea and growth stunting? It appears that the answer is yes, but results are not consistent with such a hypothesis.
- 5) Without knowing exactly what particular correlate of host protection we are looking for, there is only intuitive leap towards saying that vaccinated macaques were better protected against diarrhea and stunting. Such observation is perhaps correct (as suggested by results) but it needs to be corroborated further.

Reviewer #2 (Remarks to the Author):

General:

This is an intriguing study demonstrating that immunization with a killed whole cell C. coli vaccine induces partially protective immunity and leads to a serologic response in rhesus monkeys, and, more importantly, is associated with reduced mortality. The intervention is consistent with human data suggesting (by association) a role for Campylobacter in infant stunting in low- and middle-income countries.

The strengths are the well written text and the potential application of the intervention to an enormous problem in human child health. The weaknesses lie chiefly in the omission of important details on methodology and data.

Major comments:

1. Line 411: what was the rationale for the shift in vaccine preparation?
2. How is Campylobacter-attributable diarrhea and specific-cause hospitalization defined, in situations in which Shigella and Cryptosporidiosis are also common?
3. Lines 231-236: were the measurers blinded to the treatment?
4. Lines 234 and following: Are these differences statistically significant?
5. Line 406: Campylobacter vaccination. How are batch to batch immunogenicity differences quantified?
6. Details on the Campylobacter EIA are sparse, in this paper and in the reference in which it was initially published (Quintel, et al). How much batch to batch variability is there in the antigen preparation for the EIA? A whole bacterial killed preparation could be difficult to standardize for reproducible data, though it is possible that the same stock as previously described was used (in which case it would be good to provide data showing confidence in the reproducibility of the assay with new batches). Moreover, the flagellin preparation is no doubt enriched for flagellin, but Quintel, et al, show only that the antigen is present on a western blot, and not that other antigens are absent.
7. In human studies, there is considerable attention paid to validation of measurements of stature, but there is no description of the protocols used.
8. Do you have weights in addition to the dorsal length data?

9. The effect of vaccination on intestinal pathology is not provided, though the importance of the enteropathy is inferred in the text, and validates the use of the model. Even if there is no difference between the vaccinated and the control animals, that would be informative.

10. How was the subset of animals used for dorsal length measurements chosen?

11. Because the authors have 16s data from all groups at multiple timepoints (as shown in Figure 1) and from the mothers and infants, it would be worthwhile to answer several questions relating to the presence of *Campylobacter* species:

a. Do vaccinated mothers have any changes in their continuing colonization with *Campylobacter* that might affect the transmission to their infants?

b. Any differences in the microbiota overall of *Campylobacter*-negative infants and positive infants?

c. How does colonization change over time in infants negative at 1 month for *Campylobacter*? Do they remain negative or do they become positive later?

d. Are infants who get diarrhea those who were initially colonized or those who remained negative and became colonized later? In other words, does negative colonization predict diarrhea at all?

12. Maternal serum antibodies are more appropriately termed maternal, and not t0. It would also be helpful to display maternal antibodies independently and compare them to appreciate changes in maternal antibodies from vaccination that will influence the infants through passive transfer. It would also be interesting to see *Campylobacter* IgA concentrations in serum and breast milk, if available.

13. The catch up growth by 12 months (fig 5) seems to weaken the authors' contention that early-life *Campylobacter* infection has a lasting effect on growth. Does this catch-up growth continue? Can differences be observed in height outside of the neonatal period for macaques infected with *Campylobacter* compared to those that had not been infected? If data on subsequent growth are not available, at least this limitation deserves mention in the Discussion.

Minor comments:

1. Were p-values two-tailed?

2. Lines 181-196: it should be emphasized that these values represent animals hospitalized with diarrhea, not all diarrhea. This, however, well described elsewhere in the text and figures.

3. Line 430: what is the rationale for not absorbing the sera with *Shigella*?

4. Do you have weights in addition to the dorsal length data?

5. Was there a reduction in diarrhea associated with non-coli *Campylobacter* compared to all other causes diarrhea as hypothesized by the authors on line 224? They discuss impact on deaths but not hospitalizations caused by other species of bacteria.

6. Figure 1 should better emphasize that 2018 is the control for VI 1/3/5 and 2017 the control for VDI and VI 1/3/12.

7. In Figure 4, were there any differences between the vaccine groups in all-cause diarrhea?
8. In which vaccination groups were the 2 deaths?

Reviewer #3 (Remarks to the Author):

Review of Campylobacter vaccination reduces diarrheal disease and infant growth stunting among rhesus macaques

This study examines the effect of 3 different regimens for vaccinating infant rhesus macaques against Campylobacter coli compared to unvaccinated controls.

What are the noteworthy results?

The noteworthy results are the lack of impact of vaccination against C. coli on the development of the captive rhesus macaque gut microbiome through 12 months of age, a decrease in all-cause diarrhea-associated mortality in captive rhesus macaques through 18 months of age from vaccination against C. coli, and a transient improvement in length-to-age Z scores at 9 months of age in captive rhesus macaques vaccinated against C. coli.

Will the work be of significance to the field and related fields? How does it compare to the established literature? If the work is not original, please provide relevant references.

The work will be significant to the Campylobacter vaccine field and the related enteropathogenic bacteria vaccine field. This study is an extension of previous work done by this group on the vaccination against C. coli in captive adult rhesus macaques. In addition, this study is one of the first to demonstrate that vaccination of captive infant rhesus macaques against C. coli may impact all-cause diarrhea-associated mortality and linear infant growth.

Does the work support the conclusions and claims, or is additional evidence needed?

In general, the work partially supports the conclusions and claims. Unfortunately, the work suffers from the limited sample sizes of the vaccination groups. As a result, only one or two selected comparisons achieve significance in support of each conclusion or claim. Given the limited sample sizes, many comparisons lack significance limiting the support for the conclusions and claims.

Are there any flaws in the data analysis, interpretation and conclusions? Do these prohibit publication or require revision?

Aside from the limited sample sizes, the data analysis, interpretation, and conclusions are fine. Suggest including the actual numbers in the text, figures, and tables. A discussion on the limitations of the data and considerations for future studies would enhance the value of this paper.

Is the methodology sound? Does the work meet the expected standards in your field?

The methodology is sound. Although the work meets the expected standards in the field, additional detail and explanations would be helpful.

Is there enough detail provided in the methods for the work to be reproduced?

Yes, enough detail is provided in the methods for the work to be reproduced. This group has also published extensive work in this and related areas in captive rhesus macaques.

Minor Issues:

p. 2. Abstract: Line 1: Consider "... estimated to be responsible for ..." or something similar

Line 9: Consider "... results suggest that ..."

Line 10: Consider "... but potentially serves as an effective ..."

p. 3. Text: Line 1: Consider "...study estimated that over 500,000 ..."

Line 7: Consider "... spp. were estimated to be associated ..."

p. 4. Line 10: Suggest adding actual numbers to "76%"

p. 5. Results: Lines 15-17: Consider adding actual numbers to "84%", "76%", "79%", and "86%". The actual numbers are not included in Table 1, so they are unavailable to the reader.

p. 6. Line 28: Please check if "12,500 EU" is included in Fig. 2A

p. 8. Line 23: Please check if "Fig. 3B" should read "Fig. 4B"

p. 11. Discussion: Lines 5-6: Consider "... Campylobacter itself is estimated to be responsible for ..."

p. 22. Figure Legends: Lines 2-6: Suggest expanding the explanation in legend for Fig. 1 to provide the reader with enough information to interpret the associated figure.

pp. 27-28. Figures: Consider adding actual numbers of macaques to Fig. 3. And Fig. 4.

REVIEWER COMMENTS

Reviewer #1 (Remarks to the Author):

This manuscript represents an attempt to show that *Campylobacter* vaccine can reduce diarrhea and growth-stunting in captive juvenile rhesus macaques. This is very timely and human health translatable topic as *Campylobacter*-associated morbidity is significant in both human infants and juvenile (captive) macaques. Before manuscript can be accepted for publication, several specific issues need to be addressed.

Specific Comments:

1) Results indicate that vast majority of *Campylobacter* isolates in macaques are linked to *C. coli*. From the past results generated by other groups, there appears to be higher morbidity caused by *C. jejuni* than *C. coli* strains.

Respectfully disagree. Each Primate Center differs in terms of their local burden of enteric pathogens but at the ONPRC, *C. coli* is the most common cause of diarrhea (59% of diarrheal cases), followed by *Shigella* (12% of diarrheal cases) and *C. jejuni* (5.9% of diarrheal cases) (Quintel et al., Science Advances 2020;6:eab4511).

This indicates that the incidence of *C. coli*-associated diarrhea is approximately 10-fold greater than the incidence of *C. jejuni*-associated diarrhea observed among rhesus macaques housed at the ONPRC.

Did authors try to determine what particular *Campylobacter* strain is exclusively or more associated with clinical disease in macaques?

Agree. This is an important point and we have provided the following text at the beginning of the Results section on page 5 of the revised manuscript:

“The study involved infant macaques born in 2017 and 2018 and both LOS (lipopolysaccharide) and CPS (capsular polysaccharide) loci of the circulating strains of C. coli sequenced in this time frame (2015-2018) did not match the C. coli vaccine strain (isolated in 2013) and therefore these experiments represent a robust heterologous challenge model for measuring vaccine-mediated protection against Campylobacter-associated enteric disease²⁷”

Reference #27 is Quintel et al., Science Advances 2020;6:eab4511.

Vaccine appears to be directed broadly against all *Campylobacter*s. More specifically, it would be very informative to identify the components of *Campylobacter* that are functioning as antigenic elicitors of protective immunity.

Agree – in part. Although we agree that it would be interesting to know the specific component(s) involved with protective immunity within our *Campylobacter* vaccine, this information is not necessary for a vaccine to be successful, or for it to be FDA-approved for clinical use. For example, we don't know the specific antigen(s) presented by the whole-cell pertussis vaccine that are the most protective against whooping cough and even with the FDA-approved acellular pertussis vaccine, we don't know if one, two, three, or even four vaccine antigens are each absolutely required for protection and there is still no consensus on what the putative protective threshold is for any one of the individual pertussis antigens.

FDA approval of successful vaccines without knowledge of the specific protective antigens is not unique to bacteria either; the specific protective epitopes have not been formally determined for most of our current live virus vaccines either. For example, live-attenuated varicella zoster virus (VZV) vaccines were FDA-approved for varicella and shingles, despite no knowledge of the protective mechanism, the specific protective antigens, or whether protection is mediated by antibodies, T cells, or some combination of both humoral and cell-mediated immunity.

2) Since parasitic and viral pathogens as well as food sensitivities were all identified as potential contributors to clinical diarrheas in juvenile macaques, was there any attempt to determine such %s (even if minor) at ONPRC?

Agree – in part. We discussed the potential for food sensitivities with the ONPRC veterinarians and according to them, food allergies are not an issue at this primate center. This point of view is in alignment with prior publications (Xu et al., Clin Immunol 2013) indicating that food sensitivity in the form of celiac disease is very rare (~1% of the population) at the Tulane primate center. Moreover, we were reminded that since infant macaques are almost exclusively breastfed for up to 6 months of age or longer, it is believed to be extremely unlikely for them to develop a food allergy/sensitivity during early infancy. In contrast, and as noted in the original introduction, prior studies have shown that rhesus macaques at the ONPRC are endemic for multiple enteric pathogens including *Campylobacter*, *Shigella*, *Cryptosporidium*, and *Giardia*. All cases of diarrhea reported at the ONPRC were independently screened by the ONPRC Microbiology Core laboratory for *Campylobacter* (*C. coli* and *C. jejuni*), *Shigella* (*S. flexneri* and *S. dysenteriae*), and *Salmonella*. If the animal was not diagnosed with one of these enteric pathogens, then they were categorized as diarrhea of “unknown etiology”, which means that it could be due to other bacterial, parasitic, or viral pathogens. In the revised manuscript (page 9), we have now provided the percentage of diarrhea-associated deaths that were due to unknown etiology (e.g., 57% among unvaccinated animals).

3) In results, authors state that vaccination did not affect diversity of gut microbiome. However, it is not clear what type of diversity (alpha or beta) they are referring to. It would be informative to clarify this.

Agree. In the revised manuscript (page 6), we have now clarified that neither alpha diversity nor beta diversity were affected by *Campylobacter* vaccination.

“No difference was observed in terms of microbial alpha diversity, as measured by observed amplicon sequence variant (ASVs), at any timepoint after vaccination between the 2017 vaccinated groups (VDI-1/3/12 and VI-1/3/12) and their corresponding unvaccinated 2017 Controls or the 2018 vaccinated group (VI-1/3/5) and their contemporaneous unvaccinated 2018 Controls ($P > 0.05$, Fig. 1A). Additionally, the overall community composition (i.e., beta diversity), calculated as weighted UniFrac distance, did not differ between vaccinated groups and their contemporaneous Controls at any timepoint ($P > 0.05$, Fig. 1B, C).”

4) What was the rationale of focusing on *Campylobacter* serum IgG? Did authors anticipate this metric to correlate with protection against diarrhea and growth stunting? It appears that the answer is yes, but results are not consistent with such a hypothesis.

Agree – in part. We focused on *Campylobacter* serum IgG because the vaccine was administered intramuscularly and based on this route of vaccination, we anticipated that transplacental transfer of maternal IgG and transudation of serum IgG to sites of enteric infection/inflammation would be the most likely mechanisms of protection.

These results are consistent with the hypothesis that vaccination may reduce diarrheal disease. For example, in Fig. 2 the VDI-1/3/12 group (i.e., vaccinated dams and infants) had the highest *Campylobacter*-specific serum IgG at the 1 month time point (4,560 EU) compared to VI-1/3/12 or the VI-1/3/5 groups (628 EU and 628 EU, respectively) and the VDI-1/3/12 group showed the highest protection in the per-protocol analysis (Fig. 3) at 79% VE compared to 41% VE and 53% VE in the other vaccine groups, respectively. Likewise, if we focus only on the 12 month ELISA titers, the unvaccinated infants developed *Campylobacter*-specific antibody titers of 6,730 EU compared to 34,300 EU (VDI-1/3/12), 28,900 EU (VI-1/3/12), and 34,500 EU (VI-1/3/5) among the vaccinated cohorts and these increased IgG titers were associated with reduced diarrhea (per-protocol) with 79% VE, 41% VE, and 53% VE in these respective vaccine cohorts.

We also provide new data in the revised manuscript showing *Campylobacter*-specific IgA titers in breast milk (Supplementary Fig. 2) and compared *Campylobacter*-specific IgA titers in milk to infant BMI (Supplementary Fig. 4C). There was no significant correlation between milk IgA titers and infant BMI ($P = 0.14$). In contrast, there was a significant correlation between infant macaque serum IgG titers to *Campylobacter* and BMI (Supplementary Fig. 4D, $P = 0.01$). These results would be consistent with the hypothesis that high antibacterial serum IgG titers may be associated with improved infant outcomes.

5) Without knowing exactly what particular correlate of host protection we are looking for, there is only intuitive leap towards saying that vaccinated macaques were better protected against diarrhea and stunting. Such observation is perhaps correct (as suggested by results) but it needs to be corroborated further.

Respectfully disagree. From a historical perspective, there was no intuitive leap involved with the scientific consensus that smallpox vaccination resulted in the complete eradication of Variola virus despite the fact that we still do not know the specific antigen (or antigens – indeed, there are 73 proteins in the intact virus) involved with protection and a correlate of immunity has never been formally determined. Likewise, the *Campylobacter* vaccination study described here provides an evidence-based approach showing that immunization against this specific enteric pathogen results in statistically significant protection against *C. coli*-associated diarrhea and infant growth stunting by directly comparing vaccinated vs. unvaccinated infant macaques in a robust model of natural exposure. This current study was not designed to determine the correlate of immunity, as this may be best determined in a direct-challenge model in which the timing and dose of exposure are known and can be compared to the pre-existing immune response at the time of challenge, preferably with and without depletion of B cell or T cell responses in a manner similar to determining the correlates of immunity to monkeypox in NHP (Edghill-Smith et al., Nat Med 2005;11:740). Although these types of studies are outside of the scope of this current paper on vaccine-mediated protection against natural endemic *Campylobacter* exposure, these points have been added to the Discussion in the revised manuscript (page 14).

Reviewer #2 (Remarks to the Author):

General:

This is an intriguing study demonstrating that immunization with a killed whole cell *C. coli* vaccine induces partially protective immunity and leads to a serologic response in rhesus monkeys, and, more importantly, is associated with reduced mortality. The intervention is consistent with human data suggesting (by association) a role for *Campylobacter* in infant stunting in low- and middle-income countries.

The strengths are the well written text and the potential application of the intervention to an enormous problem in human child health. The weaknesses lie chiefly in the omission of important details on methodology and data.

Major comments:

1. Line 411: what was the rationale for the shift in vaccine preparation?

Agree. Between 2017 and 2018, we found a publication showing that brief exposure to formaldehyde can enhance bacterial protein stability (e.g., pertussis toxin) and subsequent vaccine-induced immunity (Rappuoli et al., Vaccine 1994;12:579) and so the vaccine preparation was modified to see if this might increase protective immunity following *Campylobacter* vaccination in the 2018 birth cohort. This information and reference have now been added to the Methods in the revised manuscript.

2. How is *Campylobacter*-attributable diarrhea and specific-cause hospitalization defined, in situations in which *Shigella* and *Cryptosporidiosis* are also common?

Agree. At the time that an infant is hospitalized for diarrhea, a rectal swab is taken and independently evaluated by direct microbial culture on selective media at the ONPRC clinical pathology laboratory and screened for *C. coli*, *C. jejuni*, *S. flexneri*, *S. dysenteriae*, and *Salmonella* spp. The ONPRC does not routinely screen for enteric parasites such as *Cryptosporidium* or *Giardia* and so cases of diarrhea associated with these (or other) enteric pathogens are typically listed as “diarrhea of unknown etiology”. This information has been clarified in the Methods section of the revised manuscript.

3. Lines 231-236: were the measurers blinded to the treatment?

Agree. The measurers were not blinded to the treatment group but they were unaware of the prior length measurement of an animal at the time of each new longitudinal measurement. This meant that each time they performed a dorsal length measurement, they did not know if the change in length was large or small compared to the prior measurement for that particular animal. This information has now been added to the Methods of the revised manuscript along with more detail regarding how the dorsal length measurements were performed.

4. Lines 234 and following: Are these differences statistically significant?

Agree. On page 11, we have added the descriptive P values for the raw dorsal length scores and indicated that, to follow the gold standard in the field and for translatable effect size to human data, we further examined LAZ scores, and inferential statistics were performed using regression models to adjust for known confounding variables such as initial length at 1 month of age, sex, and non-*C. coli* diarrhea.

5. Line 406: Campylobacter vaccination. How are batch to batch immunogenicity differences quantified?
Agree. The same vaccine strain of *Campylobacter* was grown in the same medium, at the same temperature, for the same time prior to harvest/purification/inactivation. Likewise, the protein concentration of each vaccine lot was determined and diluted to a final concentration of 40 micrograms/dose. The following comparison of batch-specific immunogenicity (2017 vs. 2018 vaccine lots) have been provided in the Methods of the revised manuscript:

“The immunogenicity of vaccine lots prepared in 2017 and 2018 appeared to be similar since Campylobacter-specific IgG titers among vaccinated infants in each birth cohort were not significantly different at 3 months of age [i.e., 2 months post-primary vaccination (Fig. 2A); geometric mean: 37,400 EU versus 38,200 EU, in 2017 vs. 2018, respectively, P = 0.50, Student’s t-test]. Similar results were also observed when measuring C. coli flagellin-specific responses at 2 months after primary vaccination (Supplemental Fig. 1A; geometric mean: 15,600 EU vs. 14,700 EU, in 2017 vs. 2018, respectively, P = 0.85, Student’s t-test).”

6. Details on the Campylobacter EIA are sparse, in this paper and in the reference in which it was initially published (Quintel, et al). How much batch to batch variability is there in the antigen preparation for the EIA? A whole bacterial killed preparation could be difficult to standardize for reproducible data, though it is possible that the same stock as previously described was used (in which case it would be good to provide data showing confidence in the reproducibility of the assay with new batches). Moreover, the flagellin preparation is no doubt enriched for flagellin, but Quintel, et al, show only that the antigen is present on a western blot, and not that other antigens are absent.

Agree – in part. We agree that more experimental detail should be provided in the manuscript but there may be some confusion over the data described in the Quintel et al. study. This paper provided an SDS-PAGE of a *Campylobacter* whole-cell lysate and showed that a whole-cell lysate contained many proteins including flagellin (confirmed by Western blot; Quintel et al. Fig. 5D), but there was no SDS-PAGE of purified flagellin included in that paper. In terms of the *Campylobacter* flagellin ELISA, we adapted a previously published approach for preparing flagellin antigen that shows high purity (Newell et al., J Gen Microbiol 1984;130:1201) and have added this reference to the revised manuscript. The original Methods described this as “semi-purified flagellin” which we believe is in line with the Reviewer’s perspective that this antigen is enriched for flagellin antigen compared to a bacterial whole-cell lysate.

There may also be some confusion over the type of *Campylobacter* antigen used in our experiments. We did not use inactivated whole bacteria to coat the ELISA plates, but instead we used an inactivated whole bacterial cell lysate, as indicated in the original manuscript and in the prior study by Quintel et al.

We believe that perhaps the major concern here is the potential variability between ELISA assays performed with different lots of bacterial antigen on different days? We agree that this is an important aspect in these studies and over the past 25 years of ELISA-based antibody research across multiple infection models, we have

mitigated this potential issue by using an in-house serum standard that is run in duplicate on every ELISA plate and we use this to normalize ELISA titers across time, between plates within the same experiment, between plates assayed in different experiments, between different operators, and between different antigen preparations. For example: Let's assume that a *Campylobacter* ELISA has been optimized for sensitivity and specificity and the in-house standard scores 10,000 ELISA units (EU). Next, let's assume that one year later, a new batch of ELISA plates were prepared with a new lot of *Campylobacter* antigen and the in-house standard no longer scores 10,000, but instead it scores 11,000. In order to compare the data from the new batch of ELISA plates to the older ones prepared/performed one year earlier, we use a coefficient based on the in-house standard that is locked in to a specific ELISA titer. In this example, the new ELISA plates would be normalized by multiplying the in-house standard score (as well as all of the other serum titers from this particular ELISA plate) by 0.909 ($10,000/11,000 = 0.909$). After multiplying by this co-efficient, the new titer for the in-house standard is once again 10,000 ($11,000 \times 0.909$) and now it matches the results obtained from one year earlier. By using this normalization approach, the other serum samples on that same ELISA plate will likewise be consistent with the results obtained in earlier experiments (or in future experiments).

In terms of the ELISA protocol used in this study, we have now provided a more detailed description of this assay and how it was optimized in the revised manuscript:

“ELISA plates were coated with an optimized concentration of HydroVax-inactivated Campylobacter whole-cell lysate (1 ug/mL) or semi-purified flagellin antigen (1 ug/mL) as previously described²⁷ except that serum samples were not preadsorbed with Shigella prior to performing the assay since this did not appear to alter assay performance. The concentration of each Campylobacter antigen was optimized in small-scale pilot studies using serial dilutions of protein to provide the highest sensitivity with the lowest background. The Campylobacter whole-cell lysate was prepared from late log-phase C. coli (strain: NTIC13) grown in suspension overnight and homogenized in PBS (Fisher PowerGen 125 homogenator, setting 5 with homogenization performed 3 x 60 seconds/each with cooling on ice in between) to prepare a bacterial lysate that was aliquoted and stored at -80°C. To prepare semi-purified flagellin antigen, an approach was adapted from Newell et al.⁶⁰ in which the homogenized lysate was centrifuged twice at 2000xg for 30 min, with the pellet resuspended in PBS each time. The resuspended pellet was then processed by ultracentrifugation for 3 hours at 100,000xg. Semi-purified flagellin pellets were resuspended in PBS, aliquoted, and stored at -80°C. Once the optimized concentration of antigen was determined, ELISA plates were coated overnight at 2-8 °C in bulk (i.e., 20-80 ELISA plates/batch) and then stored at -20°C until use. Plates were thawed and unbound antigen was removed before plates were blocked with 5% non-fat dry milk in PBS-T (PBS supplemented with 0.05% Tween-20) for 1 hour at room temperature. Plates were rinsed 1X with PBS-T and incubated for 1 hour with 3-fold serial dilutions of RM breast milk or heat-inactivated serum. Fifty microliters of 10% H₂O₂ was added (3% H₂O₂, final concentration) to each well and incubated for 30 minutes at room temperature to inactivate potential blood-borne pathogens. Plates were washed 3X with PBS-T and incubated with an optimized dilution (1:2,000) of a mouse anti-rhesus macaque IgG-HRP antibody (clone SB108a, Southern Biotech) or an optimized dilution of 1:1,000 of a goat anti-rhesus macaque IgA-HRP antibody (orb21435, Biorbyt) for 1 hour at room temperature. Plates were rinsed 3X with PBS-T and 100 µl of o-phenylenediamine dihydrochloride (OPD, 0.04%) substrate containing 0.01% H₂O₂ in citrate buffer (0.05M citric acid, 0.1M Na₂HPO₄, pH = 5.0) was added for 20 min, with colorimetric development stopped by the addition of an equal volume of 1M HCl. Optical densities (OD) were measured at 490 nm and a log-log transformation of the linear range of OD (0.05 to 1.5 OD) versus reciprocal serum dilution was performed and end-point ELISA titers were determined as the reciprocal of the serum dilution needed to reach an OD = 0.10. Each ELISA plate included a Campylobacter-immune serum standard that was serially 3-fold diluted in duplicate on each plate to allow normalization between ELISA plates performed in the same experiment or between experiments performed on different days or with ELISA plates coated with different production lots of antigen. Each experimental serum sample was tested at least in duplicate and paired samples with >25% coefficient of variation (CV) were repeated. The operator

was blinded in terms of group designation at the time they were performing the assay. The Campylobacter-immune serum IgG and IgA serum standard was from a rhesus macaque obtained at 5.7 years after recovery from 6 prior C. coli-associated hospitalizations for diarrhea.”

7. In human studies, there is considerable attention paid to validation of measurements of stature, but there is no description of the protocols used.

Agree. We apologize for not providing more detailed description of how the measurements were performed in the original manuscript and have now added this information to the Methods of the revised manuscript. *“Dorsal length measurements were performed using a Seca 210 measuring mat developed for measuring human infants and toddlers (Seca, product number 210 1721 004). Following ketamine or Telazol anesthesia of the infant, two individuals work together to perform the measurements. The infant was placed on the measuring mat on its back with its head in the center of the head positioner. The legs were then fully extended with the toes pointed up before locking the foot positioner in place and reading the dorsal length in centimeters. Although the measurers were not blinded to the treatment group, they were unaware of the prior length measurement of an animal at the time of each new longitudinal measurement. All procedures, including dorsal length measurements, weight measurements, vaccinations, blood draws and rectal swabs were performed under ketamine or Telazol anesthesia by trained personnel under the supervision of veterinary staff.”*

8. Do you have weights in addition to the dorsal length data?

Agree. We measured the weights of the longitudinal animals at each time point where the dorsal lengths were determined and we have now provided this information in a new Supplemental Fig. 4 in addition to new data regarding BMI vs. serum K/T ratios in a new Fig. 6 provided in the revised manuscript.

9. The effect of vaccination on intestinal pathology is not provided, though the importance of the enteropathy is inferred in the text, and validates the use of the model. Even if there is no difference between the vaccinated and the control animals, that would be informative.

Agree – in part. In the Introduction of the original manuscript, we described the results of our recent enteric histopathology paper that showed essentially 100% of infant rhesus macaques have histological abnormalities in the small intestine that are consistent with enteropathy/EED, regardless of their growth rates. However, we realize here that we didn’t make it clear in the original introduction that when infant macaques were stratified according to growth trajectories, the degree of histopathology in the small intestine was not different between healthy vs. growth-faltering animals. Instead, the most significant differences between enteric histopathology and infant growth rates were identified in the large intestine, which we hypothesize is due to the colon/large intestine functioning as an energy salvage organ when nutritional absorption in the small intestine is compromised (Norsa et al., Am J Clin Nutr 2019;109:1112).

When we initiated these *Campylobacter* vaccination studies in 2017, we had not yet analysed the histological data from the 2022 Hendrickson Nature Communications paper on EED. Although the original IACUC protocol for the *Campylobacter* vaccination study included having two animals/group brought to necropsy, we didn’t know at the time that this would be insufficient for making valid comparisons between groups. Based on the 2022 Hendrickson study, we now know that there is a range of histological abnormalities among healthy vs. growth faltering infants and that 7-10 animals/group were required to identify statistically significant differences in histopathology in different locations in the small or large intestine. Moreover, in this current *Campylobacter* vaccination study, we didn’t know that 9 months of age was the peak difference in dorsal length kinetics until we had reached the 12 month time point. Therefore, we missed the window of opportunity for comparing infant groups at the point of the largest differences in growth velocity. With only two 12-month old animals/group that were brought to necropsy (the last time point stipulated in the original IACUC protocol for dorsal length measurements), we don’t have information regarding histological differences at the 9 month time point. We have not reviewed the slides with our collaborators for histological analysis but we have concerns

that the number of samples ($n = 2$) are simply too few to provide meaningful interpretation. This will require repeating the study at some point in the future with more animals and performing blinded histological analysis with sufficient numbers of samples at the appropriate time points (e.g., 9 months of age) to not be skewed by small sample size or by “catch-up growth” which is observed by 12 months of age in these studies as well as in human infant studies at a similar biological time point (see Discussion page 17-18). The question of small sample size is especially problematic with certain site assessments such as the duodenum which can be highly variable. For instance, we mentioned this in the Discussion of the Hendrickson Nature Commun 2020 study, “*The lack of significant differences in small intestine pathology appears to be further supported by recent clinical studies involving histological assessment of the small intestine of growth-stunted children that also did not identify a significant association between duodenal biopsy scores and anthropometric parameters including height-for-age Z-scores or weight-for-height standard deviation Z-scores³⁵. In other words, children with varying degrees of growth stunting could not be distinguished from each other based on histological analysis of the small intestine.*”

However, we agree with the Reviewer that this is an interesting question and, in the discussion, we comment on the observation that animals with high K/T ratios have more severe growth stunting and in previous studies, infant macaques with the highest K/T ratios had higher rates of histological abnormalities in the large intestine. We further emphasize that this as an important area for future investigation in the Discussion (page 18).

10. How was the subset of animals used for dorsal length measurements chosen?

The subset of infants followed longitudinally in each group were chosen based on their health (no major injuries or broken bones at 1 month of age) and the hierarchy of their dams as well as their infant-rearing history in consultation with ONPRC animal husbandry staff who were not involved in the study and did not know whether the animals would be in a vaccination group or a control group. In addition, although the longitudinal infants were not formally randomized, they were typically chosen to be included in a vaccine or control group prior to their first physical exam and their length and weight were not known until after they had been enrolled within a particular study group. This information has been added to the Methods in the revised manuscript. While there was no formal randomization performed, the initial average dorsal lengths differed by only 0.1 cm at the time of inclusion in the study (34.2 cm for vaccinated infants vs. 34.1 cm for unvaccinated infants) indicating that there was little to no skewing of infant length at the time of their initial group assignments.

11. Because the authors have 16s data from all groups at multiple timepoints (as shown in Figure 1) and from the mothers and infants, it would be worthwhile to answer several questions relating to the presence of Campylobacter species:

a. Do vaccinated mothers have any changes in their continuing colonization with Campylobacter that might affect the transmission to their infants?

Agree. The colonization status of the vaccinated dams did not appear to impact the rates of *C. coli* colonization of their infants at 1 month of age since 76% of the infants born to vaccinated dams in the VDI-1/3/12 group were colonized with *C. coli* by 1 month of age and this was not significantly different than the frequency of *C. coli*-positive infants (79%) in the VI-1/3/12 group that were born to unvaccinated dams in the same birth year ($P=0.99$, Fisher’s Exact Test). Moreover, of the 6 infants in the VDI-1/3/12 group that were *C. coli*-negative at 1 month of age, 2 infants were born to *C. coli*-negative dams and 4 were born to *C. coli*-positive dams, indicating that there was no direct relationship between the colonization status of the vaccinated dams and their infants at the 1-month time point. This information has been added to page 5 of the revised manuscript.

b. Any differences in the microbiota overall of Campylobacter-negative infants and positive infants?

Agree – in part. Although it would be interesting to determine the potential difference in microbiota between *Campylobacter*-positive and *Campylobacter*-negative infants, unfortunately there are too few infants that remain *Campylobacter*-negative to allow for statistically valid comparisons to be made. Although there were between 4-7 infants/group that were *Campylobacter*-negative at 1 month of age (Table 1), by 3 months of age,

there was only 1 *Campylobacter*-negative infant in the Control group, 1 *Campylobacter*-negative infant in the VDI-1/3/12 group, 1 *Campylobacter*-negative infant in the VI-1/3/12 group and 2 *Campylobacter*-negative infants in the VI-1/3/5 group. Valid microbiome comparisons cannot be made based on a single subject/group and together, this indicates that we would not have sufficient statistical power to compare such low numbers of culture-negative infants to culture-positive infants.

c. How does colonization change over time in infants negative at 1 month for *Campylobacter*? Do they remain negative or do they become positive later?

Agree. We have updated Table 1 and it shows that between 76-84% of infants were colonized with *C. coli* by 1 month of age and that 100% of the animals (in all groups) are colonized with *C. coli* by the 12-month time point. Colonization occurs at an early age and we have now added more information to the Table 1 legend indicating that 96% (24/25), 96% (23/24), 96% (26/27), and 96% (48/50) of the infant macaques in the Control, VDI-1/3/12, VI-1/3/12, or VI-1/3/5 cohorts were colonized with *C. coli* by the time that they were 3 months old.

d. Are infants who get diarrhea those who were initially colonized or those who remained negative and became colonized later? In other words, does negative colonization predict diarrhea at all?

Agree – in part. We agree that although it is possible that an extended delay in *Campylobacter* colonization could reduce the future risk of infant diarrhea and be predictive of improved outcomes, the early and high prevalence of colonization among infant rhesus macaques in this model (96% *C. coli* colonization by 3 months of age) and the small number of *C. coli*-negative infants at 1 month (1M) of age together makes it difficult to reliably address this question because we are not sufficiently powered to perform this analysis. In these studies, being *C. coli*-negative at 1 month of age did not appear to predict future risk of *C. coli* diarrhea since 4/21 (19%) of the Control infants that were *C. coli*-positive at 1 month of age were subsequently hospitalized with *C. coli* diarrhea whereas 2/4 (50%) of Control infants that were *C. coli*-negative at 1 month also were eventually hospitalized with *C. coli* diarrhea. This information has been added to pages 5-6 in the revised manuscript.

12. Maternal serum antibodies are more appropriately termed maternal, and not t0. It would also be helpful to display maternal antibodies independently and compare them to appreciate changes in maternal antibodies from vaccination that will influence the infants through passive transfer. It would also be interesting to see *Campylobacter* IgA concentrations in serum and breast milk, if available.

Agree – in part. We agree that maternal antibodies should be clearly marked and we have replaced t0 with an “M” in Fig. 2A and Supplementary Fig. 1A and indicated in the figure legends that these are the maternal antibody titers of the dams. Likewise, we now display the maternal antibodies independently in Fig. 2B and Supplementary Fig. 1B for comparison with their respective infant antibody titers. *Campylobacter*-specific serum IgA titers were not measured because, unlike serum IgG, serum IgA is not actively transported through the placenta to the fetus. Likewise, although serum IgG is actively taken up by FcRn-positive epithelial cells in the small intestine and delivered to the lumen of the gut, monomeric serum IgA is not actively transported to the gut by this mechanism (Horton and Vidarsson Front Immunol 2013;4;200:1). We have added this information to the Discussion in the revised manuscript (pages 14-15) along with a reference indicating that the majority of intestinal IgG is serum transudate or bile-derived whereas $\geq 98\%$ of intestinal IgA is of the secretory type and is not believed to be serum-derived (Meckelein et al., Clin Diagn. Lab. Immunol. 2003;10:831-834). In addition, we have added a paper showing that passive intravenous transfer of serum IgG in pigtailed macaques was protective against an enteric viral pathogen, rotavirus, indicating that transudation of serum IgG can provide protection at mucosal sites in the gut (Westerman et al., PNAS 2005;102:7268).

Breastmilk is difficult to obtain from rhesus macaques but samples were available from 7-10 dams from the VDI-1/3/12, VD-1/3/12, and unvaccinated Control groups at the 1 month and 3 month time points and new figures have been added to the revised manuscript showing the levels of *Campylobacter*-specific IgA in breastmilk (Supplementary Fig. 2A and 2B) and their relationship to infant BMI (Supplementary Fig. 4C).

13. The catch up growth by 12 months (fig 5) seems to weaken the authors' contention that early-life *Campylobacter* infection has a lasting effect on growth. Does this catch-up growth continue?

Respectfully disagree. This is an important question and we believe that the smaller (yet still statistically significant) difference in growth rates observed among vaccinated infant macaques at 12 months of age is not a detriment to this model but instead it may actually improve the reliability of this model for mimicking human disease because it more closely mirrors the results and catch-up growth that is observed among growth-stunted human children. In the original manuscript, we noted this in the Discussion (in bold italics below) but we have now also provided additional information (shown in yellow highlighting in the second paragraph below) to more clearly emphasize this point:

*Another advantage of the rhesus macaque model is that these animals develop and mature at approximately 3-4-fold faster rates than humans, reaching sexual maturity by ~4 years of age with an average lifespan of approximately 25 years when raised in captivity⁴⁴. Based on a 4-fold faster maturation rate compared to humans, this indicates that a 9-month old rhesus macaque would be developmentally similar to a 2.5 year old child, a 1 year old macaque would be similar to a 4 year old child and monitoring diarrheal mortality for up to 18 months of follow-up would be similar to following children to 6 years of age. Interestingly, we found the greatest vaccine-associated improvement in linear growth stunting at 9 months of age (2.7% difference, Tukey adjusted, $P=0.001$) and although the differences were smaller by 12 months of age, they were still statistically significant (0.7% difference, Tukey adjusted, $P=0.025$). **Similar observations of catch-up growth among growth stunted children in a MAL-ED cohort study have also been identified by 5 years of age³², indicating that infant macaques appear to follow similar patterns of early linear growth stunting followed by later catch-up growth as that observed among human children.** Since diarrhea-associated mortality rates are highest among children <5 years of age, monitoring infant macaques for 12-18 months provides a more rapid assessment of experimental intervention success (or failure) in an expedited time frame in addition to providing an opportunity to measure gut histology/inflammation in more detail than can be performed during most clinical studies²⁰.*

In response to this comment and to better clarify the comparisons between the infant macaque model and the latest results on human infant growth stunting/catch-up growth, we have modified the text to read as follows: ***“Similar observations of catch-up growth among growth stunted children in a MAL-ED cohort study have also been identified by 5 years of age³³. Indeed, among the children who were stunted at 24 months (n = 426), 185 (43%) were no longer stunted at 60 months. This suggests that the reduced difference infant macaque growth stunting observed at a similar stage of pediatric development (i.e., 1 year of age for macaques vs. 4-5 years of age for humans) appears to be mimicking a similar pattern of early linear growth stunting followed by later catch-up growth.”***

The dorsal length measurements were performed until 12 months of age, after which the animals were released from the study protocol and no linear growth information is available beyond this time point.

Can differences be observed in height outside of the neonatal period for macaques infected with *Campylobacter* compared to those that had not been infected? If data on subsequent growth are not available, at least this limitation deserves mention in the Discussion.

Agree. We agree that this is an interesting question, but it is not feasible to address this particular topic in the current rhesus macaque model because nearly all of the infants (96%) are colonized with *Campylobacter* sometime between 1 to 3 months of age and 100% of the infants are colonized on or before 12 months of age. This means that it is not possible to compare the linear growth rate kinetics of *Campylobacter*-infected animals to non-infected animals because this latter group does not exist under these hyper-endemic exposure conditions and this information has been added as a limitation in the Discussion (page 14).

Minor comments:

1. Were p-values two-tailed?

Agree. All P values were two-tailed where applicable and this information has been added to the Methods in the revised manuscript.

2. Lines 181-196: it should be emphasized that these values represent animals hospitalized with diarrhea, not all diarrhea. This, however, well described elsewhere in the text and figures.

Agree. We have modified this introductory sentence to now read, “*Diarrheal disease incidence/hospitalization and mortality was monitored...*” and we have emphasized that this study focuses on animals that were hospitalized with diarrhea in this section of the text and throughout the revised manuscript.

3. Line 430: what is the rationale for not absorbing the sera with Shigella?

Agree. In the earlier study by Quintel et al., sera were absorbed against *Shigella* as a precaution to remove potential cross-reactive antibodies. However, when we compared pre-adsorbed and non-pre-adsorbed sera in subsequent ELISA assays, we saw no difference in antibody binding profiles and so we removed this step since it did not impact the outcome of the experiments.

4. Do you have weights in addition to the dorsal length data?

Agree. We have provided weight and BMI assessments in the revised manuscript (New Fig. 6B and Supplemental Fig. 4).

5. Was there a reduction in diarrhea associated with non-coli Campylobacter compared to all other causes diarrhea as hypothesized by the authors on line 224? They discuss impact on deaths but not hospitalizations caused by other species of bacteria.

Agree. Vaccine efficacy against all-cause diarrhea-associated hospitalization (per-protocol) was 14% (P=0.71) for VDI-1/3/12, not estimable (P=0.61) for VI-1/3/12, and 54% (P=0.04) for VI-1/3/5 animals. After combining all vaccinated infant macaques, VE = 27% (P=0.18). This suggested an overall trend in lower all-cause diarrhea-associated hospitalizations but this was only statistically significant among infants in the VI-1/3/5 group. This information has been added to the Results (page 9) in the revised manuscript.

6. Figure 1 should better emphasize that 2018 is the control for VI 1/3/5 and 2017 the control for VDI and VI 1/3/12.

Agree. We have modified the labels in Figure 1 to indicate the birth year for each infant cohort.

7. In Figure 4, were there any differences between the vaccine groups in all-cause diarrhea?

Agree. Please see the answer to Question 5 (above) and page 9 in the revised manuscript.

8. In which vaccination groups were the 2 deaths?

Agree. There was 1 non-*C. coli*-associated diarrhea death in the VDI-1/3/12 group and 1 non-*C. coli*-associated diarrhea death in the VI-1/3/12 group and this information has been added to page 9 in the revised manuscript.

Reviewer #3 (Remarks to the Author):

Review of Campylobacter vaccination reduces diarrheal disease and infant growth stunting among rhesus macaques

This study examines the effect of 3 different regimens for vaccinating infant rhesus macaques against Campylobacter coli compared to unvaccinated controls.

What are the noteworthy results?

The noteworthy results are the lack of impact of vaccination against *C. coli* on the development of the

captive rhesus macaque gut microbiome through 12 months of age, a decrease in all-cause diarrhea-associated mortality in captive rhesus macaques through 18 months of age from vaccination against *C. coli*, and a transient improvement in length-to-age Z scores at 9 months of age in captive rhesus macaques vaccinated against *C. coli*.

Will the work be of significance to the field and related fields? How does it compare to the established literature? If the work is not original, please provide relevant references.

The work will be significant to the *Campylobacter* vaccine field and the related enteropathogenic bacteria vaccine field. This study is an extension of previous work done by this group on the vaccination against *C. coli* in captive adult rhesus macaques. In addition, this study is one of the first to demonstrate that vaccination of captive infant rhesus macaques against *C. coli* may impact all-cause diarrhea-associated mortality and linear infant growth.

Does the work support the conclusions and claims, or is additional evidence needed?

In general, the work partially supports the conclusions and claims. Unfortunately, the work suffers from the limited sample sizes of the vaccination groups. As a result, only one or two selected comparisons achieve significance in support of each conclusion or claim. Given the limited sample sizes, many comparisons lack significance limiting the support for the conclusions and claims.

Are there any flaws in the data analysis, interpretation and conclusions? Do these prohibit publication or require revision?

Aside from the limited sample sizes, the data analysis, interpretation, and conclusions are fine. Suggest including the actual numbers in the text, figures, and tables. A discussion on the limitations of the data and considerations for future studies would enhance the value of this paper.

Agree. In the revised manuscript, we provide either individual symbols for each animal or have added the animal numbers/group to the text, figures, figure legends and tables as well as provided a section in the Discussion regarding the limitations of the dataset and considerations for future studies. Importantly, we provide substantial new data showing a) statistically significant inverse correlations between infant dorsal length and serum K/T ratios (new Fig. 6A), b) significant inverse correlations between infant BMI and serum K/T ratios (new Fig. 6B), c) highly significant correlations between dorsal length and weight for all age groups (Supplementary Fig. 4A), d) significantly increased BMI among infants in the VDI-1/3/12 group compared to VI-1/3/12 or controls (Supplementary Fig. 4B), and e) a significant positive correlation between high *Campylobacter*-specific serum IgG levels and higher infant BMI at 1 month of age (Supplementary Fig. 4D).

Is the methodology sound? Does the work meet the expected standards in your field?

The methodology is sound. Although the work meets the expected standards in the field, additional detail and explanations would be helpful.

Agree. In the revised manuscript, we have provided additional details (e.g., animal numbers to figures/table) and more complete information in the Methods and Results sections regarding animal measurements, ELISA protocols, in-house ELISA standards, etc.

Is there enough detail provided in the methods for the work to be reproduced?

Yes, enough detail is provided in the methods for the work to be reproduced. This group has also published extensive work in this and related areas in captive rhesus macaques.

Minor Issues:

p. 2. Abstract: Line 1: Consider "... estimated to be responsible for ..." or something similar

Agree. We have updated the Abstract to make these changes in the revised manuscript.

Line 9: Consider “... results suggest that ...”

Respectfully disagree. We began to update the text to make these requested changes in the revised manuscript but upon review of Nature Communications Formatting Instructions regarding the Abstract we realized that: “The final sentence must begin with a phrase like “In this work” or “Here, we show”....

We realized that in order to follow journal style, the final sentence had to be updated and it has now been changed from, “Together these results indicate...” to “Here, we show....” We hope that this is considered acceptable.

Line 10: Consider “... but potentially serves as an effective ...”

Agree. We have updated the text to make these changes in the revised manuscript.

p. 3. Text: Line 1: Consider “...study estimated that over 500,000 ...”

Agree. We have updated the text to make these changes in the revised manuscript.

Line 7: Consider “ ... spp. were estimated to be associated ...”

Agree. We have modified the text in the revised manuscript to state, “In 2010, it was estimated that *Campylobacter* spp. were associated...”.

p. 4. Line 10: Suggest adding actual numbers to “76%”

Agree. We have changed the text to include the specific number of animals in these calculations:

“We found that the rates of Campylobacter-associated diarrhea were reduced among vaccinated infants, with no Campylobacter-associated deaths recorded (0/90 vs. 7/248 C. coli deaths among unvaccinated infants). All-cause diarrhea-associated mortality was also reduced by 76% (2/90 among vaccinated infants vs. 23/248 all-cause diarrhea deaths among unvaccinated infants, P = 0.03).”

p. 5. Results: Lines 15-17: Consider adding actual numbers to “84%”, “76%”, “79%”, and “86%”. The actual numbers are not included in Table 1, so they are unavailable to the reader.

Agree. We have updated the text to include the number of animals in addition to the percentages both in the main text as well as in Table 1.

p. 6. Line 28: Please check if “12,500 EU” is included in Fig. 2A

Agree. In the original manuscript, the data for the maternal antibodies could be seen in Fig. 2A but the titer, 12,500 EU, was only provided in the text describing the figure and was not written in the figure itself since none of the antibody titers in Fig. 2A had numerical scores listed with them. In the revised manuscript, we have expanded Fig. 2B in response to Reviewer #1’s comments (see above) to include the maternal antibodies in Fig. 2B. In this revised figure, the titer is no longer 12,500 EU (i.e., the ELISA titer for all dams included in the project) but is now listed as 11,100 EU since this number is based only on the dams who had corresponding infants that survived to 1 year of age for inclusion in the figure.

p. 8. Line 23: Please check if “Fig. 3B” should read “Fig. 4B”

Agree. Thank you for identifying this typographical error – it has now been corrected to read “Fig. 4B”.

p. 11. Discussion: Lines 5-6: Consider “... Campylobacter itself is estimated to be responsible for ...”

Agree. We have updated the text to make these changes in the revised manuscript.

p. 22. Figure Legends: Lines 2-6: Suggest expanding the explanation in legend for Fig. 1 to provide the reader with enough information to interpret the associated figure.

Agree. We have included additional text in the figure legend to improve the readability and interpretation of Figure 1.

pp. 27-28. Figures: Consider adding actual numbers of macaques to Fig. 3. And Fig. 4.
Agree. We have provided the number of animals in each group to Fig. 3 and Fig. 4.

REVIEWERS' COMMENTS

Reviewer #1 (Remarks to the Author):

Current version of the manuscript was substantially improved. There is still one question that was not addressed by authors. Question # 2 in my original review is referring to viral and parasitic agents, not to gluten sensitivity. Please address this question appropriately.

Reviewer #2 (Remarks to the Author):

The authors have satisfactorily addressed each of our comments, though we wish to encourage expanded response to major comment 9. We accept the authors' explanation that any sort of a statistical analysis would be futile in view of the low numbers. Nonetheless, this cohort is so unique and small bowel biopsies are so rare in the enteropathy literature, that we believe it would be appropriate to include the biopsies (ideally as whole slide images) in the supplemental materials section. The authors can certainly insert a caveat that the numbers are too few to draw conclusions.

Response to Reviewer comments on revised manuscript

Reviewer #1:

Current version of the manuscript was substantially improved. There is still one question that was not addressed by authors. Question # 2 in my original review is referring to viral and parasitic agents, not to gluten sensitivity. Please address this question appropriately.

Agree. We apologize for any misunderstandings regarding our previous response.

We mentioned in the prior response to reviewer comments that;

“All cases of diarrhea reported at the ONPRC were independently screened by the ONPRC Microbiology Core laboratory for *Campylobacter* (*C. coli* and *C. jejuni*), *Shigella* (*S. flexneri* and *S. dysenteriae*), and *Salmonella*. If the animal was not diagnosed with one of these enteric pathogens, then they were categorized as diarrhea of “unknown etiology”, which means that it could be due to other bacterial, parasitic, or viral pathogens.”

We should have further clarified that the ONPRC only screens for these four specific pathogens and does not routinely screen for any other bacterial, viral, or parasitic pathogens. We also did not perform any additional pathogen screening because we were limited to only testing animals that were directly enrolled/assigned to our vaccination studies and we did not have access or approvals to screen the rest of the infants in the colony that were monitored passively for diarrheal incidence through the ONPRC electronic health monitoring system and this is why we relied upon the results provided by the ONPRC Microbiology Core Laboratory. Together, this means that when animals were hospitalized with diarrhea of unknown etiology, it is possible that other bacterial, viral or parasitic infections may have played a role but there is no information on what may have been the root cause in these cases that were negative for *Campylobacter*, *Shigella* or *Salmonella*.

The text in the Methods section of the revised manuscript has been updated to the following:

Rhesus macaques had rectal swabs taken at each visit in addition to when infants were hospitalized for diarrhea and these were evaluated in a blinded manner by direct microbial culture on selective media at the ONPRC clinical pathology laboratory which routinely screens for *C. coli*, *C. jejuni*, *S. flexneri*, *S. dysenteriae*, and *Salmonella* spp. If the sample was negative for these specific enteric pathogens, then it is possible that other bacterial, viral or parasitic pathogens may have been present but without formal identification/confirmation, the samples were coded as “normal flora” or “diarrhea of unknown etiology”.

Reviewer #2:

The authors have satisfactorily addressed each of our comments, though we wish to encourage expanded response to major comment 9. We accept the authors’ explanation that any sort of a statistical analysis would be futile in view of the low numbers. Nonetheless, this cohort is so unique and small bowel biopsies are so rare in the enteropathy literature, that we believe it would be appropriate to include the biopsies (ideally as whole slide images) in the supplemental materials section. The authors can certainly insert a caveat that the numbers are too few to draw conclusions.

Agree – in part. In terms of this current NHP vaccination, immunogenicity, and protection study, we hope to reiterate that it was not designed for histological analysis and here we wish to further emphasize that there are no histological samples available that were obtained during the most relevant period of observation (1-12 months of age) in which the greatest differences in infant growth stunting were identified.

In the prior response to reviewer’s comments, I was mistaken when I mentioned the availability of histology slides – we do not have any H&E slides from the animals studied in this current manuscript. Upon closer review of our tissue block inventory that can be used to prepare H&E slides, we realized that the 2 animals from the VI-1/3/12 cohort were brought to necropsy at 16 months of age and the 2 Control animals from 2017 were brought to necropsy at 15 months of age (not 12 months as previously stated). This is substantially outside of the age range where differences in growth stunting were observed (3-12 months, Fig. 5) and we do not have any growth kinetics data on infants that are this far beyond 12 months of age. Unfortunately, there were no

necropsies performed for the VDI-1/3/12 cohort, meaning that there are no tissue blocks available for any of the animals at any time point from the most important vaccine cohort since the VDI-1/3/12 group represented the only cohort that demonstrated a statistically significant improvement in infant growth stunting.

There are other caveats to consider as well; for instance, only one of the VI-1/3/12 macaques with archived tissue blocks belonged in the longitudinal cohort (Fig. 5) whereas the other animal was not followed longitudinally and therefore the information on its growth rates is limited and there is no information on its K/T ratios. In summary, we only have tissue from a single vaccinated macaque that was measured longitudinally (albeit following the inferior VI-1/3/12 vaccination schedule) and this animal was brought to necropsy at 16 months of age (i.e., equivalent to a 5 year old child). In retrospect, and in comparison with older human pediatric cohorts (S.A. Richard et al. BMC Public Health 21, 1246 2021), this time point is less likely to show characteristics of prior growth stunting due to the catch-up growth that occurs in older age groups. As noted by Richard et al., 43% of growth-stunted 2 year old children no longer showed signs/evidence of growth stunting by 5 years of age. Therefore, analysis of 15-16 month old rhesus macaques that are maturationally equivalent to 5 year old children, are not relevant to the current study of young infant growth stunting.

We agree with the Reviewer that small bowel biopsies are rare in the enteropathy literature and the rhesus macaque model provides a unique opportunity to study clinically-relevant gut histopathology in the context of enteric infection. However, in contrast to the *Campylobacter* vaccine protection study described here that is missing appropriate tissue sample collection within the observed timeframe of infant growth stunting, we respectfully refer the Reviewer to our recent study that addresses these points (Hendrickson et al., Nat Commun 2022). This 2022 study provides a comprehensive histological examination of gut pathology among healthy vs. growth-faltering infant rhesus macaques and we believe that this particular histology-focused publication has the information that the Reviewer appears to be looking for: it includes graded histological samples of the entire GI tract (including duodenum, jejunum, proximal/mid/distal ileum, cecum, ascending/transverse/descending colon, and rectum) involving a sizeable number of young (6-11 month old) infant macaques (n = 17 macaques, including 7 healthy infants for comparison with 10 growth-faltering infants). Small bowel histopathology was provided directly in the 2022 Nat Commun publication (Fig. 2) and ~170 digitized whole slide images/tissue sections of the GI tract of both healthy and growth faltering infant macaques (i.e., 10 gut tissues per animal) have been uploaded to the Washington University Digital Pathology Exchange (WUPAX) and are currently available to the public. Based on the lack of appropriate sample collection for the purposes of histology in this current paper, in addition to access to a well-defined, fully characterized, and publicly available histopathology resource assessing the entire infant rhesus macaque gastrointestinal tract (Hendrickson et al., 2022), we do not believe that the limited number of tissue blocks that we have on hand will provide meaningful information to the research community in its current form. As described in the revised Discussion of the current manuscript, we hope to perform further in-depth analysis of *Campylobacter* vaccination of infant macaques and, based on what we have learned to date, we can perform appropriately controlled histological analysis with group sizes that will allow one to determine if there is an association between gut histopathology of the large or small intestine with infant growth stunting.